# Identifying Good and Bad Neurons for Task-Level Controllable LLMs

## Abstract

Despite their remarkable capabilities, the complex mechanisms by which neurons influence Large Language Models (LLMs) remain opaque, posing significant challenges for understanding and steering LLMs. While recent studies made progress on identifying responsible neurons for certain abilities, these ability-specific methods are infeasible for task-focused scenarios requiring coordinated use of multiple abilities. Moreover, these approaches focus only on supportive neurons accounting for target performance while neglecting neurons with other roles, *e.g.*, inhibitive roles, resulting in an incomplete view of LLMs in task execution. Also, they are often customized for specific data structures, lacking flexibility for diverse tasks with varying input-output formats. To address these challenges, we propose **NeuronLLM**, a novel task-level LLM understanding framework that adopts the biological principle of *functional antagonism* for LLM neuron identification, with the key insight that task performance is jointly determined by neurons with two opposing roles: "good" neurons that facilitate task completion and "bad" neurons that inhibit it. NeuronLLM is instantiated by two main modules: *Question-Answering-based Task Transformation (**QATT**)* and *Contrastive Neuron Identification (**CNI**)*. QATT transforms diverse tasks into unified question-answering format, enabling NeuronLLM to understand LLMs under different tasks; CNI identifies good and bad neurons via a new cross-entropy-based contrastive scoring method, featuring a holistic view of neuron analysis. Comprehensive experiments on LLMs of different sizes and families show that NeuronLLM substantially outperforms existing methods in identifying task-relevant neurons across four NLP tasks, providing new insights into LLM functional organization.

## 1 Introduction

Large language models (LLMs) have demonstrated impressive generalization abilities and are known to encode a wide range of knowledge and capabilities (Yuan et al., 2023). Despite these remarkable performance, our understanding of their internal mechanisms remains limited, posing an important issue about interpretability, trust, and mitigation (Singh et al., 2024). Taking an analogy to the brain of biology sense, where various components tend to specialize in different cognitive abilities (Bari & Robbins, 2013), AI researchers find that such functional differentiation could also appear in the components of LLMs (Xiao et al., 2024), *e.g.*, in their latent feature space (Zou et al., 2025) or their projection heads (Olsson et al., 2022). Despite the success of these methods, more fine-grained understanding of the LLMs, such as at the neuron level, remains an essential but under-explored problem, having significant applications in different use cases of controllable LLMs. For example, hunting for neurons that are tied to a specific capability or behavior, *e.g.*, truthfulness, repetition, and safety, allows us to mitigate the issues in this specific aspect of LLMs (Hiraoka & Inui, 2024; Chen et al., 2024; Li et al., 2025). Although effective, these existing LLM neuron identification methods are limited to single capabilities. They become infeasible for steering LLMs in task-focused application scenarios. This is because i) completing a task typically requires a constellation of various abilities; ii) accurately decomposing all possible abilities required for a task is very difficult, if not impossible (Elhage et al., 2022; Yax et al., 2023), *e.g.*, LLM-based models for stock price prediction would rely on many underlying capabilities, such as comprehension of financial statements and news, macroeconomic indicator analysis, global market interdependency analysis,

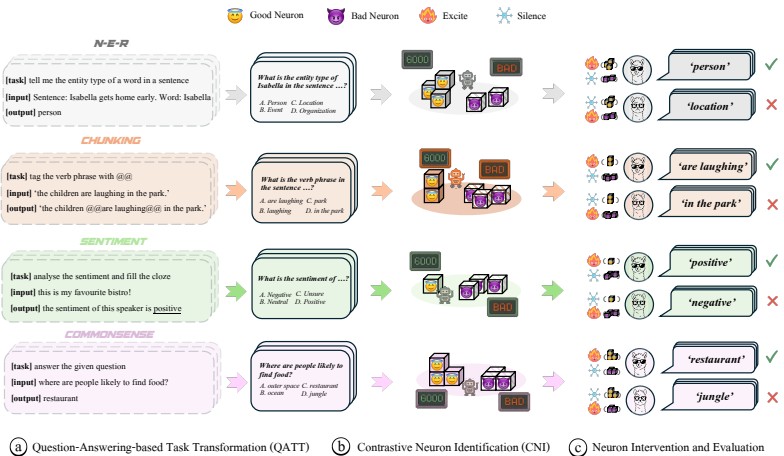

Figure 1: Overview of NeuronLLM. It first transforms diverse tasks into a QA-based task format, on which a cross-entropy-based neuron attribution method is devised to identify good and bad neurons for task-level steering of LLMs.

logical reasoning, etc; and iii) one would need to apply the corresponding attribution method for each ability, if such a method exists.

To fill this gap, in this work we step back and explore the problem of identifying a small set of neurons for understanding and controlling LLMs at the task level as a whole, which could be viewed as a top-down philosophy in the sense of Hopfieldian perspective from cognitive neuroscience (Hopfield, 1982). Although less intricate than capability-level understanding, task-level LLM understanding is also challenging. First, within the black-box architecture of billion-parameter LLMs, the complex mechanisms by which different neurons interact to determine task performance remain largely unknown. Although very recent neuron-localization approaches show promising results for understanding such mechanisms, they only focus on finding the supportive neurons that account for certain target performance, leaving neurons with other potential roles neglected (Li et al., 2025). This results in an incomplete, isolated view of the complex mechanisms that govern task execution (Bertalanffy, 1968; Anderson, 1972). Second, different tasks (*e.g.*, tagging, classification, open-ended generation) often exhibit diverse input-output formats, but existing methods are typically customized for specific data formats, lacking the flexibility needed for diverse task scenarios (Dai et al., 2022).

To address these challenges, we propose **NeuronLLM**, a novel framework that leverages neurons of two opposing roles: good and bad—those being supportive and inhibitory respectively for a given task—for a holistic steering of LLMs at the task level. A key insight in NeuronLLM is that the performance of LLMs in completing a task is determined not only by the good neurons but also the bad neurons and their interaction with the good ones, as shown in Fig.1(b). This idea is inspired by *functional antagonism*, a well-established principle in biology-related disciplines (Lu, 2021; Demertzi et al., 2022; Fu et al., 2023; Rocha et al., 2023), which indicates that a task completion (*e.g.*, basal ganglia's motor circuits) is featured by a "direct" pathway (*i.e.*, a group of neurons) in our brain that facilitates the completion and an "indirect" pathway that suppresses it; and the coordinated interaction of both pathways together endows the full process, *e.g.*, human subjects with healthy motor control (Rocha et al., 2023).

NeuronLLM is a generic framework that consists of two main modules, including a *Question-Answering-based Task Transformation (**QATT**)* module and a *Contrastive Neuron Identification (**CNI**)* module. To guarantee the generalizability to different tasks, QATT is devised to transform diverse tasks into a unified Question-Answering (QA) form, as shown in Fig.1(a). This enables the subsequent neuron attribution method to model consistent output targets and generalize across different task types, while preserving essential characteristics of the original task, thereby facilitating a task-agnostic interpretation ability in NeuronLLM. A new neuron attribution method is further introduced in CNI to enable an accurate cross-entropy-based contrastive analysis of the importance of LLM neurons. Furthermore, we show that different existing neuron attribution methods can be incorporated into the CNI module to achieve improved task-level controllable LLMs. Our main contributions are summarized as follows:

- We propose **NeuronLLM**, a novel framework that reveals the existence of neurons with opposing roles (good and bad) in LLMs for holistic task-level understanding and steering. To our best knowledge, NeuronLLM is the first framework to adopt the idea of *functional antagonism* from biology into neuron identification inside LLMs: task performance is jointly determined by both supportive and inhibitory neurons and their coordinated interaction. This enables more accurate identification of task-relevant neurons and provides new insights into the functional organization of LLMs.

- We introduce two key modules, **QATT** and **CNI**, to instantiate NeuronLLM. QATT offers an effective way to transform diverse tasks to a universal multi-choice QA form, enabling task-agnostic neuron attribution in NeuronLLM while preserving essential task characteristics. Building upon this transformation, CNI proposes a new cross-entropy-based contrastive neuron scoring method that is naturally suited for the QA format yielded by QATT, providing an accurate measurement of neuron importance w.r.t. a given task. Additionally, CNI is designed to be flexible, allowing existing or future attribution methods to be integrated for improved performance.

- Extensive results on LLaMA 2-7B, Baichuan 2-7B, and LLaMA 2-13B show that Neuron-LLM substantially outperforms state-of-the-art methods over multiple NLP tasks.

## 2 RELATED WORK

### 2.1 FUNCTIONAL ANTAGONISM IN BIOLOGY

Examples of opposing role specialization of components in complex systems and their coordinated interaction can be broadly found in biology-related disciplines: silencing a small set of striatal interneurons dismantles stereotyped habits (O'Hare et al., 2017); lesions to the lateral habenula improve working memory in hemiparkinsonian rats (Du et al., 2018; Cardoso-Cruz et al., 2025); activating "PV" neurons in mouse's visual cortex reduces its visual contrast sensitivity (Del Rosario et al., 2025); and deliberately suppressing competing processes can enhance cognition—an "addition-by-subtraction" mechanism exploited in rehabilitative therapy (Luber & Lisanby, 2014). Such role specialization also varies with task context: the prefrontal cortex supports logical control yet hampers creativity when overactive (Chrysikou et al., 2013; Weber et al., 2022). No studies on exploring such roles in LLMs have been reported.

### 2.2 INTERPRETABILITY OF NEURAL NETWORKS

Early interpretability research focused on conventional deep neural networks, such as backprop-based visualization methods (Simonyan et al., 2014; Zeiler & Fergus, 2014; Nguyen et al., 2016), masking-based causal attribution (Fong & Vedaldi, 2017), surrogate-based LIME (Ribeiro et al., 2016), gradient-based grad-CAM (Selvaraju et al., 2020), and many other methods like SHAP (Lundberg & Lee, 2017).

As model complexity increased, especially with the advent of LLMs, interpretability techniques have likewise evolved (Calderon & Reichart, 2025). A notable example is the discovery of induction heads in Transformer networks, which seeks "circuits" of components (Wang et al., 2022; Olsson et al., 2022). Other methods look at representation subspaces (Geiger et al., 2024; Zou et al., 2025), generalizable patterns of information flow (Geva et al., 2023), and direction-based probes (*e.g.*, via sparse dictionary learning) for vectors that can be explained as coherent concepts or features (Huben et al., 2023; Bricken et al., 2023; Todd et al., 2024; Tigges et al., 2024; Brinkmann et al., 2025). Despite these advances, the quest to identify and interpret individual neurons remains central, partly because neurons are a natural basis for explaining network behaviors, and also because identifying a single "unit" responsible for a behavior is intuitively plausible. One representative work in this scope is Knowledge Neurons (Dai et al., 2022) which store particular facts (*e.g.*, the capital of France). Other works often focus on different capabilities, such as Syntactic Agreement, Word Appearance, Language-Mode, Chracter Pattern, Privacy, Toxicity Control, Truthfulness (Mueller et al., 2022; Chen et al., 2023; Wu et al., 2023; Tang et al., 2024; Gurnee et al., 2024; Suau et al., 2024; Song et al., 2024; Li et al., 2025), which can be categorized into activation-based, causal-based, and gradient-based. However, these methods focus only on effect of the good neurons, ignoring the role of the bad neurons.

## 3 THE PROPOSED NEURONLLM

### 3.1 PRELIMINARIES

To evaluate the positive and negative contribution of a neuron to task performance, gradients serve as natural tools indicating the relationship between targets and inputs, making it a fundamental basis for measuring the quality of LLM neurons (Sundararajan et al., 2017; Miglani et al., 2023). Following these studies, we can approximate the contribution of a neuron $w_i^l$ to target function $F$ using integrated gradients (IG):

$$\text{IG}(w_i^l) := \frac{\hat{w}_i^l}{m} \times \sum_{k=1}^{m} \frac{\partial F(\frac{k\hat{w}_i^l}{m})}{\partial w_i^l}, \tag{1}$$

where $w_i^l$ is the $i^{th}$ neuron in a $l^{th}$ Feed-Forward Network (FFN) layer, $\hat{w}_i^l$ is its assigned value, and $m$ is the number of steps to approximate the integral. This work is focused on neurons in the FFNs since FFNs in LLMs are found to encode meaningful features responsible for different abilities (Geva et al., 2021; Dai et al., 2022; Geva et al., 2023; Chen et al., 2024). If the neuron has a strong influence on $F$, the magnitude of the gradient will be significant, which in turn has large integration values, either positive or negative.

For LLMs, given a query $q$ (*e.g.*, *Paris is the capital of*), the target function $F$ is often set as the sum of the log probabilities of each token in the answer string $y$ (*e.g.*, *France*):

$$P(y|w_i^l, q) = \sum_{j=1}^{n} \log P(t_j|\hat{w}_i^l, q, t_1, \ldots, t_{j-1}), \tag{2}$$

where $y$ is tokenized into $n$ discrete tokens $\{t_1, t_2, \ldots, t_n\}$ (*e.g.*, ["F", "ran", "ce"]). Each $P(t_j|w_i^l, q, t_1, \ldots, t_{j-1})$ represents the conditional probability of generating token $t_i$ given the query prompt $q$ and previously generated tokens.

### 3.2 FRAMEWORK OVERVIEW

NeuronLLM is a general framework for task-relevant neuron identification in LLMs that tackles the two aforementioned issues: *incomplete view of analysis* and *inconsistent task format*. As illustrated in Figure 1, NeuronLLM consists of two key modules: Question-Answering-based Task Transformation (QATT) and Contrastive Neuron Identification (CNI), along with a Neuron Intervention and Evaluation module for validating the effectiveness of identified neurons.

QATT transforms diverse tasks into a unified QA format, enabling subsequent neuron attribution to focus on consistent output targets and allowing NeuronLLM to generalize across various tasks. Based on the unified format, CNI can then identify task-relevant neurons split into good and bad neurons, featuring a holistic analysis of the neurons. Within CNI, we propose a new neuron scoring method named Additive-Cross-Entropy (ACE) scoring, which accurately assesses each neuron's contribution to task execution, specially suited for the QATT-converted data. To evaluate the effectiveness of our identified task-relevant neurons, our Neuron Intervention and Evaluation module adopts classic silencing-excitation strategies from neuroscience, which compares how task performance changes before and after applying certain perturbations on these neurons. Below we introduce each component in detail.

### 3.3 QATT: QUESTION-ANSWERING-BASED TASK TRANSFORMATION

Existing neuron attribution methods are typically customized for their specific data formats, such as triplet facts (Dai et al., 2022) and preference pairs (Chen et al., 2024), which limits their application to broader tasks with varying input-output formats. We introduce QATT to unify arbitrary task formats into a standardized multiple-choice QA structure, thereby providing consistent output targets while preserving the essence of the original tasks. Specifically, as shown in Figure 1(a), assuming the original task $T$ consists of a series of input-output examples $\{e_1, \ldots, e_n\}$, QATT employs prompt engineering to augment it with the following components for each example.

- **Role & Rule Specification**: This part first specifies the role of the LLM, clarifying what it should do for the given task, such as analysing the sentiment of a text. Then it provides detailed rules for the LLM to follow when answering a multi-choice QA.
- **Question Stem**: This part refers to the paraphrased question stem which should catch the essence of original input and task. Examples are shown in Figure 1(a).
- **Distraction Choices**: This part provides four options including the correct choice and three distractors. The inclusion of both correct and incorrect choices is not trivial which we will explain in detail below.
- **One-Shot Demonstration**: We provide one demonstration example for each task to further leverage the LLM's in-context learning capabilities. This component serves as a template that helps the model understand the expected input-output mapping and also improve task performance. Research shows that even a single demonstration can improve LLM performance (Brown et al., 2020).

The proposed multiple-choice QA format is crucial for three main reasons. *i) Complete view of response signals from LLMs*. Unlike previous methods, such as Knowledge Neurons (Dai et al., 2022), that consider only the probability of generating the correct answers shown in Eq. 2, QATT also includes distraction choices (incorrect choices), in addition to the correct one, to receive a more complete view of response signals from LLMs. Intuitively, given the large vocabulary size of LLMs, task-relevant neurons may simultaneously contribute to both correct and incorrect choices. These distractors serve as contrastive information, enabling our next CNI module to more accurately evaluate the role of a neuron. Depending on the nature of the task, the distractors can be generated through randomly sampling from possible answers or using existing LLMs (see Appendix C.9). *ii) Consistent output format*. The fixed multi-choice targets facilitate the use of a consistent neuron scoring pipeline across query examples and tasks. This also aligns with established evaluation practices—many existing benchmarks employ multi-choice QAs to assess LLM capabilities (Hendrycks et al., 2021). *iii) Being more computationally efficient*. By constraining the model to select from single token options rather than generating the full answers, we require only singe-step token generation, avoiding the costly computation of gradients over the summed log probabilities in Eq. 2.

However, these advantages come with an inherent issue, *i.e.*, LLMs can sometimes answer questions correctly by chance rather than through genuine understanding, which can mislead our neuron attribution. To address this issue, QATT employs a robust validation mechanism by generating three proxy questions for each original input example, where the answer options are systematically shuffled while preserving the correct choice. This transformation expands the task $T$ from $n$ examples to $3 \times n$ proxy questions. The key insight is that truly task-relevant neurons should demonstrate consistent positive or negative contributions across these proxy questions, rather than exhibiting sporadic correctness due to chance.

### 3.4 CNI: CONTRASTIVE NEURON IDENTIFICATION

We further propose the CNI module to achieve a more holistic analysis of the importance of the neurons from two opposing roles. At the core of CNI is a new Additive-Cross-Entropy (**ACE**) scoring method, specifically designed to consider both positive-negative response spectrum enabled by the unified multi-choice QA format in QATT. ACE consists of the following two components, i) cross-entropy-based contrastive neuron scoring and ii) its additive reordering.

**Cross-Entropy-based Contrastive Neuron Scoring.** Since QATT has expanded every example $e$ of the original task into a proxy question set $\mathcal{Q} = \{p_1, p_2, p_3\}$ where each is transformed into a unified multiple-choice QA with four options (A, B, C, D), this standardized format enables us to leverage a key advantage: the question-answering process can be formulated as a multi-class classification problem over a fixed set of options. Motivated by this, ACE is proposed to leverage a cross-entropy-based contrastive scoring function to capture both the confidence of the LLM in the correct choice and its uncertainty about incorrect ones. The contrastive target function is defined as:

$$F(c^*|\frac{kw_i^l}{m}, p) = e^{-\text{crossentropy}(c^*|\frac{k\hat{w}_i^l}{m}, p_t)} = P(c^*|\frac{k\hat{w}_i^l}{m}, p_t), \qquad (3)$$

where $c^*$ is the letter of the correct choice of a proxy question $p$. Essentially, this target function is mathematically equivalent to the softmax probability of the correct choice against the other three

distraction choices, offering a novel yet easy way to model both the positive and negative effects of the LLM in completing a task. This way differs from the conventional target function as in Eq. 2, which considers solely the probability of generating the correct choice over the whole vocabulary, leading to wrongly identified neurons that actually increase/decrease both probabilities of correct and incorrect answers. This pitfall is also noticed in recent studies, including a concurrent work (Li et al., 2025), while Eq. 3 in ACE helps mitigate this issue (see Table 6 for empirical support).

**Additive Reordering of Contrastive Neuron Scores.** Replacing function $F$ in Eq. 1 with our cross-entropy-based target function in Eq. 3, we can get a rough estimation of the contribution of a neuron to understand a proxy question correctly. As mentioned in Section 3.3, LLMs can get a question correct by chance. We utilize a simple but effective mechanism to further refine the score, referred to as *additive reordering*, which is done by an aggregation over the roughly estimated scores for all three proxy questions in $\mathcal{Q}$. Formally, we define the refined estimation for the original example $e$ as:

$$\text{ES}_e(w_i^l) := \sum_{t=1}^{3} \frac{\hat{w}_i^l}{m} \sum_{k=1}^{m} \frac{\partial P(c^* | \frac{k \hat{w}_i^l}{m}, p_t)}{\partial w_i^l}. \tag{4}$$

We can obtain an example-level importance score for each neuron for a given example of the task $T$ via Eq. 4. To obtain task-level scores, we apply additive reordering to a set of such examples $\{e_1, \ldots, e_{tr}\}$ from the task to aggregate and obtain more accurate neuron scores. This is to ensure that the neurons we identified by CNI are not only supportive/inhibitory in getting a single question correct but also effective in the broad range of questions at the task level. Formally, given an example $e_j$, we define $\mathcal{G}_j$ and $\mathcal{B}_j$ respectively as the sets of good and bad neurons corresponding to the top and bottom $z$ neurons ranked by $\text{ES}_{e_j}$. The ambiguous neurons that appear in the good and bad sets across examples are removed. These ambiguous neurons are assigned with zero importance score. For the other neurons, we compute their ACE score as:

$$\text{ACE}(w_i^l) = \sum_{j=1}^{tr} \mathbb{I}[w_i^l \in \mathcal{G}_j \cup \mathcal{B}_j] \cdot \text{ES}_{e_j}(w_i^l), \tag{5}$$

where $\mathbb{I}$ is an indicator function, meaning that neurons that do not appear in any $\mathcal{G}_j$ and $\mathcal{B}_j$ will also receive a zero score. The final task-level neuron sets $\mathcal{G}^T$ and $\mathcal{B}^T$ are formed by selecting the top and bottom $K$ neurons based on their ACE scores.

### 3.5 NEURON INTERVENTION AND EVALUATION

To validate the effectiveness of the identified neurons, we adopt classic intervention approaches from neuroscience (Wiegert et al., 2017): given a query and the response value at a neuron $w_i^l$, we either: i) silence the neuron by zeroing out it via $w_i^l = 0$, or ii) excite the neuron by doubling its value $w_i^l = 2 \times \hat{w}_i^l$. The goal of neuron intervention here is to either enhance or degrade task performance. If neurons are correctly identified, exciting good neurons should enhance performance, while silencing them should degrade it. Unlike existing methods that ignore the bad neurons, NeuronLLM can leverage the interaction between the good and bad neurons via a joint intervention operator: **enhancer** that excites good + silences bad; **degrader** that silences good + excites bad. Evaluation of these neuron interventions would provide empirical evidence for functional antagonism inside LLM neurons. *Note that since we focus on neuron identification in this work, we use only these simple interventions to facilitate a straightforward evaluation of the identified neurons; more intricate LLM steering operation can be explored upon our insights in future work.*

## 4 EXPERIMENTS

### 4.1 TASKS AND DATASETS

To thoroughly evaluate our framework, we select the following four well-established NLP tasks, spanning from low-level lexical analysis to high-level abstract reasoning processes (see Figure 1(a) for examples of these tasks). *Named Entity Recognition (NER)* is a lexical-level task that requires identifying and classifying proper nouns (*e.g.*, locations) within a sentence. *Chunking* is a syntactic-level task that involves detecting shallow phrase structures such as noun phrases, verb phrases, and prepositional phrases. *Sentiment Classification* operates at the semantic-level, requiring the model

Table 1: RAC/RCC results (%) of NeuronLLM and competing methods across four NLP tasks. 'Deg' and 'Enh' refer to neuron intervention to purposely degrade and enhance the task performance, respectively (see Section 3.5). Larger RAC/RCC values indicate better performance in degrading/enhancing the LLMs. **Red** highlights the best performance per metric, **Blue** shows the second best, and Fail indicates that the intervention produced the opposite effect.

| | NER | | Chunking | | Sentiment | | Commonsense | | Average | |
|---|---|---|---|---|---|---|---|---|---|---|
| | Deg | Enh | Deg | Enh | Deg | Enh | Deg | Enh | Deg | Enh |
| **LLaMA 2-7B** | | | | | | | | | | |
| **NeuronLLM** | 53.3/64.0 | 25.6/46.0 | 35.2/60.0 | 7.8/4.0 | 66.9/80.0 | 24.3/46.0 | 50.3/62.0 | 8.9/28.0 | 51.4/66.5 | 16.7/31.0 |
| **TN** | 47.8/44.0 | 13.3/34.0 | 17.2/32.0 | 6.3/4.0 | 63.9/78.0 | 10.7/24.0 | 9.5/0.0 | 5.3/12.0 | 34.6/38.5 | 8.9/18.5 |
| **QRNCA** | 48.9/46.0 | 13.9/34.0 | 9.4/16.0 | 3.9/2.0 | 60.4/70.0 | 7.1/16.0 | Fail | 2.4/8.0 | 30.3/31.5 | 6.8/15.0 |
| **KN** | 23.7/20.0 | 10.1/20.0 | 9.4/18.0 | 5.5/2.0 | 16.1/12.5 | 5.7/5.0 | Fail | 2.8/7.5 | 12.8/11.4 | 6.0/8.6 |
| **ACT** | 0.0/0.0 | 0.0/0.0 | 1.0/0.0 | 0.0/0.0 | Fail | 0.0/0.0 | 0.0/0.0 | 0.0/0.0 | Fail | 0.0/0.0 |
| **RANDOM** | Fail | 0.7/0.0 | 0.0/0.0 | Fail | Fail | 2.4/5.0 | Fail | 0.7/0.0 | Fail | 0.7/1.3 |
| **Baichuan 2-7B** | | | | | | | | | | |
| **NeuronLLM** | 63.6/73.6 | 25.8/23.6 | 50.3/64.9 | 15.1/12.3 | 46.0/51.7 | 40.4/29.3 | 56.7/74.6 | 10.0/10.4 | 54.2/66.2 | 22.8/18.9 |
| **TN** | 7.2/9.7 | 12.4/13.8 | 47.2/59.6 | 8.8/10.5 | 3.7/1.7 | 11.2/1.7 | 7.0/6.0 | 1.5/4.5 | 16.3/19.3 | 8.5/7.6 |
| **QRNCA** | 2.9/2.8 | 12.4/12.5 | 47.2/59.6 | Fail | 5.6/5.2 | 9.3/1.7 | 18.9/23.9 | Fail | 18.7/22.9 | Fail |
| **KN** | 6.2/5.6 | 13.9/15.3 | 47.2/59.6 | 3.1/3.5 | 10.6/8.6 | Fail | 30.9/34.3 | Fail | 23.7/27.0 | 3.8/3.8 |
| **ACT** | Fail | 0.0/0.0 | 0.0/0.0 | 0.0/0.0 | 2.0/0.0 | Fail | 0.0/0.0 | Fail | 0.4/0.0 | Fail |
| **RANDOM** | 0.0/0.0 | 0.0/0.0 | Fail | 1.8/0.0 | 3.0/0.0 | Fail | 0.0/0.0 | 0.0/0.0 | Fail | 0.5/0.0 |
| **LLaMA 2-13B** | | | | | | | | | | |
| **NeuronLLM** | 32.6/33.3 | 10.0/6.7 | 28.8/46.7 | 15.9/11.1 | 36.6/41.8 | 2.9/0.0 | 33.8/37.9 | 8.1/10.6 | 33.0/40.0 | 9.2/7.1 |
| **TN** | Fail | 7.2/6.7 | 15.2/20.0 | 12.1/15.6 | Fail | 5.2/3.6 | 6.1/9.1 | 2.0/1.5 | 4.6/6.1 | 6.6/6.9 |
| **QRNCA** | Fail | 7.2/6.7 | 12.1/11.1 | 9.9/11.1 | Fail | 3.5/1.8 | 5.1/9.1 | 3.0/1.5 | 4.0/4.3 | 5.9/5.3 |
| **KN** | 9.1/5.3 | 8.6/5.3 | 9.9/13.3 | 7.6/8.9 | 1.2/1.8 | 5.8/7.3 | 1.5/1.5 | Fail | 5.4/5.5 | 5.8/5.0 |
| **ACT** | 0.9/0.0 | 0.9/1.3 | 0.0/0.0 | 1.5/0.0 | 0.0/0.0 | 0.6/0.0 | 0.0/0.0 | 0.0/0.0 | 0.2/0.0 | 0.8/0.3 |
| **RANDOM** | 0.0/0.0 | 0.0/0.0 | 1.5/2.2 | 2.3/0.0 | 0.0/0.0 | 0.0/0.0 | 0.0/0.0 | 0.0/0.0 | 0.4/0.6 | 0.6/0.0 |

to infer the overall sentiment expressed in a piece of text. *Commonsense Reasoning* represents the highest level of abstraction among the four tasks, which involves applying implicit real-world knowledge and reasoning over multiple concepts to arrive at the correct answer.

For each of these tasks, we select one popular dataset—Few-NERD (Ding et al., 2021), CoNLL-2000 (Tjong Kim Sang & Buchholz, 2000), SST-3 (Socher et al., 2013), and CommonsenseQA (Talmor et al., 2019)—and use samples from these datasets as the query examples. For each task, following prior studies (Chen et al., 2025), we construct one dataset consisting of few-shot (five) examples for the neuron identification (*i.e.*, $tr = 5$) and 100 examples (300 proxy QAs) to evaluate the task performance after neuron intervention. Details of these datasets are given in Appendix A.1.

### 4.2 EVALUATION METRICS

Two metrics based on the task-level LLM performance change before and after neuron intervention are used: Relative Accuracy Change (**RAC**) and Relative Comprehension Change (**RCC**) (see Appendix A.1 for the original task performance of the LLMs). RAC is defined as the relative change of an accuracy (Acc) measure: $RAC = \frac{|Acc_{original} - Acc_{intervened}|}{Acc_{original}} \times 100\%$, where $Acc$ is calculated over the transformed proxy QAs. RCC measures the change of the comprehension (Com) ability. We say the LLM understands the original question only if it can answer at least two of its three proxy QAs correctly. This helps avoid the measure being affected by cases that model gets right by chance. Formally, we define RCC as follows: $RCC = \frac{|Com_{original} - Com_{intervened}|}{|Com_{original}|} \times 100\%$, where $Com_{original/intervened}$ denotes the LLM comprehensibility before/after neuron intervention.

To examine the effectiveness of the identified neurons, we evaluate these two task performance changes when applying silencing/exciting intervention to the LLM neurons. A larger performance change (in either RAC or RCC) indicates better performance in the neuron identification, *i.e.*, silencing/exciting the task-level neurons should result in large decrease/increase in the task performance.

### 4.3 COMPETING METHODS

We compare NeuronLLM to two very recent SOTA methods: **i) TN** (Li et al., 2025), which does not consider bad neurons and uses the difference between the probability of the correct choice and the

average probability of the wrong options to specify the target function $F$; and **ii) QRNCA** (Chen et al., 2025), which also focuses on good neurons and specifies the target function using the probability of the correct answer. Since these two methods are not specially designed for task-level neuron attribution, to make a fair comparison, we equip them with our additive reordering mechanism to obtain task-level attribution. We also compare NeuronLLM with the below three relevant baselines. **i) KN** (Dai et al., 2022) calculates the neuron scores in a way similar to QRNCA, but, unlike our additive reordering, KN uses a count-based identification strategy by finding those most frequently appeared high-score neurons among the training set as the good neurons. NeuronLLM is compared with KN to show the effectiveness of our additive reordering mechanism. **ii) ACT** simply selects the neurons with high activation values, while **iii) RANDOM** select neurons from the FFNs randomly.

### 4.4 IMPLEMENTATION DETAILS

Three LLMs of different families and sizes, LLaMA 2-7B, Baichuan 2-7B and LLaMA 2-13B, are used (Touvron et al., 2023; Yang et al., 2025). To facilitate easy reproduction and minimize manual settings in all our experiments, we make the following consistent settings—the number of estimation steps: $m = 16$, the thresholds: $z = 5,000$ and $K = 100$—yielding 100 good and 100 bad neurons per task for NeuronLLM and 100 good neurons for the other methods. For fair comparison, regardless of the way we control a single neuron group or both, we stick to an intervention budget of 100 neurons: for the latter scenario, we vary the ratio of good to bad neurons from zero to one in an increment of 10%, and report the best performance among all these configurations for all methods.

### 4.5 MAIN RESULTS

**Performance in Identifying Task-Level Neurons.** The results of comparing NeuronLLM with the other methods is reported in Table 1. It is clear that, across all tasks and different LLM sizes, NeuronLLM substantially outperforms all competing methods in controlling LLMs for both degradation and enhancement of task completion. Specifically, on average, for LLaMA 2-7B, NeuronLLM achieves improvements of 16.8% in RAC and 28% in RCC in controlling LLMs for task performance degradation, and 7.8% in RAC and 12.5% in RCC in controlling LLMs for task performance enhancement, compared to the best contender TN. This improvement gets even more pronounced in Baichuan 2-7B and LLaMA 2-13B, where NeuronLLM consistently delivers the strongest neuron-control performance across both degradation and enhancement scenarios. The consistent superiority of NeuronLLM across tasks and model sizes stems from two key innovations: i) its holistic modeling of the influence of both good and bad neurons on task execution and ii) the balanced, contrastive neuron attribution to both correct and incorrect options. In contrast, existing methods such as TN, QRNCA and KN neglect the inhibitory effect of bad neurons and overlook their antagonistic interaction with the good neurons, leading to inaccurate attribution and failed control attempts in one or multiple cases. ACT and RANDOM do not show any non-trivial performance because of their oversimplified attribution strategy. In addition, all the results are obtained using a consistently small intervention budget (*i.e.*, $K = 100$ neurons), accounting for only 0.03% of total FFN neurons in LLaMA 2-7B, 0.03% in Baichuan-2-7B, and 0.02% in LLaMA 2-13B, highlighting the generalization and robustness of NeuronLLM across different tasks.

**NeuronLLM as an Enabler to Existing Neuron Scoring Methods.** Table 2 shows the results of plugging in existing neuron scoring methods into our NeuronLLM framework, in which we replace our proposed ACE scoring method with the one in TN/QRNCA. The results show that both TN and QRNCA achieve consistent performance improvements across all tasks and model sizes when enabled by our good-bad-neuron modeling framework. The gains are more substantial on Baichuan 2-7B and LLaMA 2-13B, especially for degradation. This indicates that our holistic neuron identification principle provides a generalizable framework to various neuron attribution methods. Moreover, the more pronounced improvements on larger LLMs reveal an important insight: as model complexity increases, simply focusing on supportive neurons becomes an increasingly limited strategy, probably because the functional antagonism between opposing neurons gets more intense, considering that the larger model can embed more capabilities. This makes our comprehensive approach more valuable for understanding complex neural interactions inside advanced LLMs.

Table 2: Performance of existing SOTA methods TN and QRNCA empowered by NeuronLLM.

| LLMs | TN (Deg) | | TN (Enh) | | QRNCA (Deg) | | QRNCA (Enh) | |
|---|---|---|---|---|---|---|---|---|
| | Original | Enabled | Original | Enabled | Original | Enabled | Original | Enabled |
| **LLaMA 2-7B** | 34.6/38.5 | **39.7/44.5** | 8.9/18.5 | **13.3/24.5** | 30.3/31.5 | **35.4/40.5** | 6.8/15.0 | **11.8/22.0** |
| **Baichuan 2-7B** | 16.3/19.3 | **33.4/40.2** | 8.5/7.6 | **19.0/15.0** | 18.7/22.9 | **34.3/41.9** | Fail | **14.4/9.9** |
| **LLaMA 2-13B** | 4.6/6.1 | **15.2/19.8** | 6.6/6.9 | **7.9/9.3** | 4.0/4.3 | **14.4/20.6** | 5.9/5.3 | **7.1/7.9** |

Table 3: Results of ablation on intervening good neurons, bad neurons, or both.

| Intervention | LLaMA 2-7B | | | Baichuan 2-7B | | | LLaMA 2-13B | | |
|---|---|---|---|---|---|---|---|---|---|
| | Good | Bad | Both | Good | Bad | Both | Good | Bad | Both |
| **Deg** | 42.1/48.5 | 44.6/54.5 | **51.4/66.5** | 46.0/56.0 | 28.9/35.8 | **54.2/66.2** | 22.8/25.4 | 26.0/28.1 | **33.0/39.9** |
| **Enh** | 14.3/29.5 | 10.8/22.5 | **16.7/31.0** | 19.3/15.7 | 19.3/13.4 | **22.8/18.9** | 7.9/4.8 | 3.7/3.3 | **9.2/7.1** |

## 4.6 FURTHER ANALYSIS OF NEURONLLM

**Ablation Study.** *i) Joint modeling of good and bad neurons.* Table 3 presents an ablation analysis that dissects the individual contributions of good, bad neurons, and their combined effect. The results are averaged across tasks and the full results can be found in Appendix C.3. It is clear that controlling either "Good" or "Bad" neurons individually yields substantial performance changes, demonstrating that LLMs indeed contain functionally opposing neurons—similar to biological findings where both excitatory and inhibitory units coexist to regulate system functions. Morevoer, the joint modeling in the "Both" strategy consistently outperforms the individual controls, validating our functional antagonism hypothesis in LLMs. *ii) ACE scoring.* The effectiveness of our proposed ACE scoring method can be justified when comparing the average performance of NeuronLLM in Table 1 with that of NeuronLLM-enabled TN and QRNCA in Table 2. NeuronLLM consistently outperforms the best competing method TN (enabled) across three LLMs: on average by 17% RAC and 23% RCC for Deg, and 3% RAC and 3% RCC for Enh; similar improvements are shown for QRNCA.

**Functionalities of Task-Level Neurons.** By identifying task-level neurons through Neuron-LLM, we reveal some interesting observations on the working mechanisms of LLMs. *i) Common neurons exist across tasks.* We find that we can further decompose task-level neurons. Specifically, there are some common neurons shared by the identified neuron sets for the four tasks. Intervening these common neurons can produce consistent effects across all four tasks, as shown in Table 5 in Appendix C.1, highlighting shared abilities required for different NLP tasks.

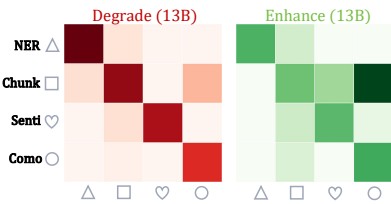

Figure 2: Cross-task evaluation.

*ii) Task-specific neurons show localized effects.* In contrast, after excluding common neurons, the remaining neurons tend to be more task-specific, which primarily affect their corresponding individual tasks only, with weaker cross-task interference, as shown by the clear diagonal on the left in Figure 2, indicating that they represent task-specific capabilities (see Appendix C.2 for more results). *iii) Enhancement vs. degradation asymmetry.* A slightly different phenomenon is observed for the enhancement . As shown in Figure 2, although we can also observe a diagonal-like trend, the enhancement of task-specific neurons sometimes improves other tasks, possibly by firing some previously weak capabilities beyond their minimal thresholds. We discuss this in greater details in Appendix C.7. *iv) Task-dependent neuron functionality*: We also find that the same neurons can be beneficial for one task but detrimental for another, aligning with biological findings where neuron contributions vary by context (see Table 8 in Appendix C.5). *v) Layer distribution of task-level neurons.* The identified neurons are predominantly located in the middle layers and the top layers as depicted in Figures 5, 6 and 7 in Appendix C.6, aligning with previous findings (Li et al., 2025).

## 5 CONCLUSION

We introduce NeuronLLM, a novel framework inspired by biological functional antagonism for task-level neuron identification in LLMs. Unlike prior methods that focus only on supportive neurons, our approach systematically considers both good and bad neurons for better identification. The proposed QATT module enables our framework generalizable across diverse task formats, while our CNI module leverages a cross-entropy-based contrastive scoring method to accurately evaluate the neuron importance for task execution. Extensive experiments with LLMs of different families and sizes show that NeuronLLM substantially outperforms state-of-the-art methods, opening new avenues for LLM interpretability and controllability.

## ETHICAL STATEMENT

All the authors of this work have read the ICLR Code of Ethics and we confirm it adheres closely to it. This research is based on publicly available open-source large language models and datasets. NeuronLLM aims to improve the interpretability and controllability of LLMs, which is a crucial step towards making these models safer and more reliable. While the ability to control model behavior could be misused, our primary goal is to provide tools for positive interventions, such as mitigating biases and enhancing factual accuracy.

## REPRODUCIBILITY STATEMENT

To ensure the reproducibility of our findings, we will release our source code, including the implementation of NeuronLLM and the experimental scripts, upon publication. All datasets used in our experiments are publicly available, and we provide detailed descriptions and preprocessing steps in Appendix A.1. Further implementation details, including the computing infrastructure, prompt templates for QATT, and hyperparameter settings, are documented in Appendix B.

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

APPENDIX

A.1 DETAILS OF SELECTED TASKS AND DATASETS

We select four distinct NLP tasks for our experiments: Named Entity Recognition (NER), Chunking, Sentiment Classification, and Commonsense Reasoning. This selection is motivated by two key considerations:

- **Linguistic Hierarchy and Functional Organization.** These four tasks represent different levels of linguistic processing: lexical (NER), syntactic (Chunking), semantic (Sentiment), as well as high-level reasoning (Commonsense). The hierarchical relationship among these tasks suggests potential complex functional associations within LLMs, including both shared general capabilities utilized across multiple tasks and specialized functions specific to individual tasks. We explore these intricate relationships, as demonstrated in Section 4.6 of the main paper.

- **Task Feasibility and Model Competence.** These tasks represent well-established problems in NLP with extensive classical datasets and evaluation protocols. LLMs trained on diverse corpora naturally acquire varying degrees of competence in these fundamental linguistic tasks, providing a solid foundation for meaningful neuron attribution. This stands in contrast to overly complex tasks where LLMs themselves fail to demonstrate adequate performance—in such cases, task-relevant neuron identification would become meaningless, as there would be no genuine specific mechanisms to localize. Consequently, we focus on tasks where the target models exhibit capability to ensure reliable neuron attribution.

For specific task configurations, we make the following choices of datasets to balance task complexity with model performance:

- **Named Entity Recognition**: We use Few-NERD (Ding et al., 2021) which is a manually annotated NER dataset drawn from English Wikipedia. It contains a hierarchical label schema comprising 8 coarse-grained and 66 fine-grained entity types. We focus on the coarse-grained classification as the latter presents substantially greater complexity that exceeds the reliable performance range of the tested models.

- **Chunking**: We create a small, simplified dataset derived from CoNLL-2000 (Tjong Kim Sang & Buchholz, 2000), as the original benchmark proves challenging for all the three models without specific finetuning (with less than 20% RAC and RCC).

- **Sentiment Classification**: We employ the popular Stanford Sentiment Treebank (SST-3) which contains annotated full sentences extracted from movie reviews. Each sentence is labeled across three sentiment categories: positive, neutral and negative. As a well-studied benchmark, SST-3 provides an appropriate level of complexity for LLMs. Since the original dataset contains only 3 categories, we use an additional option "Not Sure" as the fourth distractor.

- **Commonsense Reasoning**: We utilize CommonsenseQA (Talmor et al., 2019), which evaluates the model's ability to apply multi-hop inference and the use of background knowledge not explicitly stated in the input. Questions are crowdsourced based on ConceptNet relations to require implicit world knowledge. The multiple-choice format naturally aligns with our QATT transformation. Since there are some questions that have five options, we randomly exclude one distractor from them.

To comprehensively demonstrate the effectiveness of neuron intervention, our evaluation datasets should include both examples that the models can and cannot understand correctly. This balanced composition enables us to observe both degradation effects (when performance decreases from correct to incorrect responses) and enhancement effects (when performance improves from incorrect to correct responses) following corresponding intervention. Specifically, for each task, we sample 50 examples that LLaMA 2-7B can comprehend correctly and 50 examples that it cannot handle adequately after QATT transformation. These examples are then combined to form our evaluation set. The original performance for each task is shown in Table 4.

Table 4: Original performance of LLaMA 2-7B, Baichuan 2-7B and LLaMA 2-13B on each task (before intervention). Acc and Com represent the model's baseline capability on each task.

| **LLaMA 2-7B** | | | | |
| --- | --- | --- | --- | --- |
| Metric | NER | Chunking | Sentiment | ComSense |
| *Acc* (%) | 60 | 43 | 56 | 56 |
| *Com* (%) | 50 | 50 | 50 | 50 |
| **Baichuan 2-7B** | | | | |
| Metric | NER | Chunking | Sentiment | ComSense |
| *Acc* (%) | 70 | 53 | 54 | 67 |
| *Com* (%) | 72 | 57 | 58 | 67 |
| **LLaMA 2-13B** | | | | |
| Metric | NER | Chunking | Sentiment | ComSense |
| *Acc* (%) | 74 | 44 | 57 | 66 |
| *Com* (%) | 75 | 45 | 55 | 66 |

## B   MORE IMPLEMENTATION DETAILS

### B.1   PROMPT TEMPLATES USED IN QATT

As demonstrated in Section 3.3, QATT-transformed questions incorporate five key components: Role, Rule, Question Stem, Distraction Choices, and One-Shot Demonstration. This unified format enables the subsequent CNI module to more accurately assess neuron role. By integrating these components, the augmented questions enhance model task comprehension through the in-context learning capabilities of LLMs.

Specifically, the role, rule, and one-shot demonstration components work together to specify task requirements, define expected output formats, and provide contextual reference knowledge. Meanwhile, the question stem and distraction choices establish the multiple-choice QA format that is essential for NeuronLLM's identification, as discussed in the main text. An illustrative example of a QATT-transformed question is presented in Figure 3.

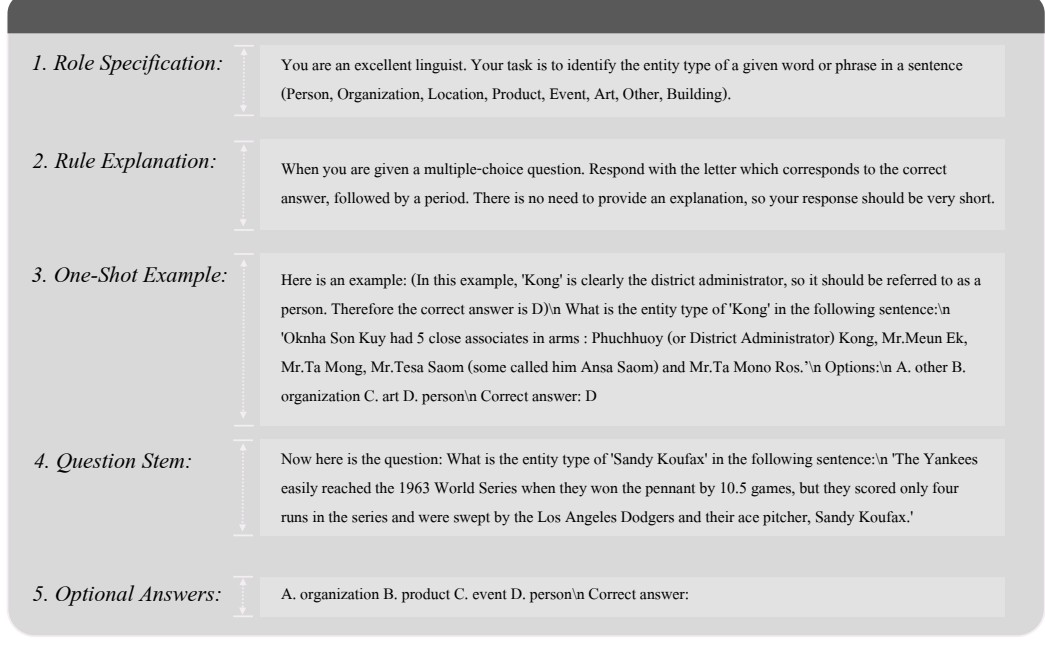

Figure 3: The example of a QATT-transformed question, consisting the components introduced in Section 3.3.

### B.2 Computing Infrastructure Used

The experiments are conducted on a Linux server with an AMD CPU (AMD EPYC 9554 64-Core Processor) and one NVIDIA H200 GPU with 141GB GPU memory. For all competing methods and NeuronLLM, the code is implemented with PyTorch 2.7.1 and Python 3.11.13.

## C Additional Empirical Results

### C.1 Common Neurons Exist across Tasks.

As mentioned in the main text, we find that we can further decompose task-level neurons into task-specific neurons and common neurons. Specifically, common good/bad neurons refers to those neurons that are detected in more than one good/bad sets of tasks. Intervening these common neurons can produce consistent effects across all four tasks, highlighting shared abilities required for different NLP tasks. Table 5 shows the impact of controlling 100 common neurons on each task for LLaMA 2-7B, Baichuan 2-7B and LLaMA 2-13B.

Table 5: The detailed impact of perturbing common ability neurons on each individual task for LLaMA 2-7B, Baichuan 2-7B and LLaMA 2-13B models.

| LLaMA 2-7B | | | | | |
|---|---|---|---|---|---|
| Intervention | NER | Chunking | Sentiment | ComSense | Average |
| Deg | 42.8/40.0 | 32.0/50.0 | 47.3/54.0 | 32.0/28.0 | **38.5/43.0** |
| Enh | 19.4/38.0 | 4.7/4.0 | 15.4/30.0 | 5.3/16.0 | **11.2/22.0** |
| Baichuan 2-7B | | | | | |
| Intervention | NER | Chunking | Sentiment | ComSense | Average |
| Deg | 53.6/62.5 | 47.2/59.6 | 49.1/74.1 | 62.19/73.1 | **53.0/67.3** |
| Enh | 13.9/13.9 | 15.1/12.3 | 34.2/24.1 | 12.4/16.4 | **18.9/16.7** |
| LLaMA 2-13B | | | | | |
| Intervention | NER | Chunking | Sentiment | ComSense | Average |
| Deg | 9.1/8.0 | 10.6/20.0 | 22.7/23.7 | 15.7/16.7 | **14.5/17.1** |
| Enh | 2.3/1.3 | 10.6/11.1 | 4.1/5.5 | 4.6/7.6 | **5.4/6.4** |

### C.2 Task-specific Neurons Show Localized Effects.

After excluding common neurons from task-relevant neurons, the remaining ones tend to be more task-specific, which primarily affect their corresponding tasks only, with weaker cross-task interference, as shown by the clear diagonal in Figure 4 below, indicating that they probably represent unique task capabilities.

### C.3 Full Ablation Analysis on NeuronLLM-enabled Methods

While the original TN and QRNCA do not consider bad neurons, we can integrate them into our framework to improve their performance (as shown in Table 2). For these NeuronLLM-enabled methods, we present a complete ablation analysis in Table 6, dissecting the individual contributions of good, bad neurons and their combined effect. Notably, NeuronLLM consistently outperforms the best competing method TN across three LLMs: by 12% RAC and 22% RCC for Deg, and 4% RAC and 7% RCC for Enh on LLaMA 2-7B; and by 21% RAC and 26% RCC for Deg, and 4% RAC and 4% RCC for Enh on Baichuan 2-7B; and by 18% RAC and 20% RCC for Deg on LLaMA 2-13B. As for Enh on the 13B model, three methods achieve comparable performance after being enabled by NeuronLLM. These results further validate the effectiveness of NeuronLLM in identifying task-relevant neurons, and the functional antagonism hypothesis that both good and bad neurons jointly determine task execution in LLMs.

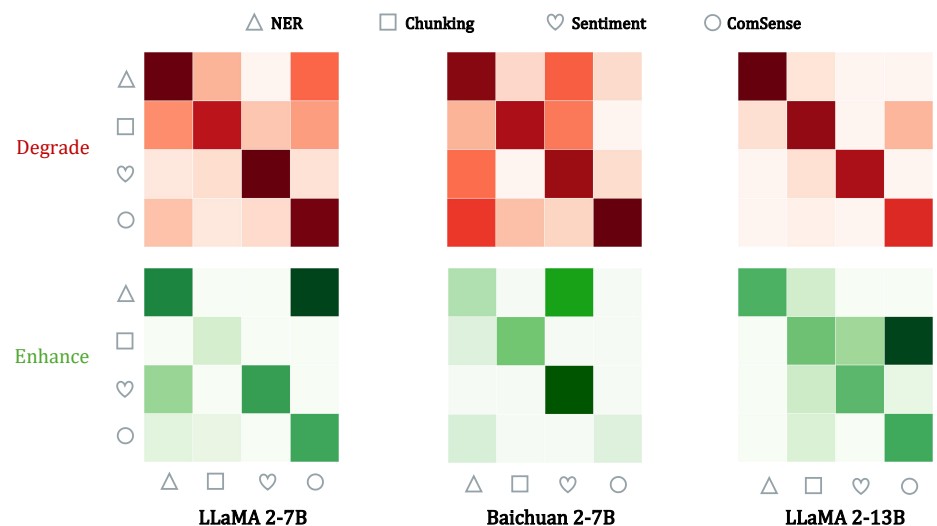

Figure 4: Evaluation of cross-task abilities of the neurons. The vertical axis represents the task data used for evaluation, while the horizontal axis indicates the task-specific neurons identified by NeuronLLM (excluding common neurons across tasks).

Table 6: Full RAC/RCC results for the ablation study. Across all tasks and model sizes, jointly controlling both "Good" and "Bad" neurons (the "Both" strategy) consistently outperforms controlling either group individually under the same intervention budget. This holds for NeuronLLM as well as NeuronLLM-enabled methods. Compared to single-group control which sometimes fails, NeuronLLM demonstrates superior robustness. These results validate our functional antagonism hypothesis in LLMs: task performance is determined by both supportive and inhibitory neurons and their coordinated interaction. Without NeuronLLM, the isolated analysis of either group is not able to catch the holistic picture of the task execution in LLMs.

### LLaMA 2-7B

| Method | Eval | NER Good | NER Bad | NER Both | Chunking Good | Chunking Bad | Chunking Both | Sentiment Good | Sentiment Bad | Sentiment Both | Commonsense Good | Commonsense Bad | Commonsense Both | AVERAGE Good | AVERAGE Bad | AVERAGE Both |
|---|---|---|---|---|---|---|---|---|---|---|---|---|---|---|---|---|
| NeuronLLM | Deg | 38.9/34.0 | 45.0/50.0 | **53.3/64.0** | 30.5/48.0 | 28.1/44.0 | **35.2/60.0** | 65.7/78.0 | 66.9/80.0 | **66.9/80.0** | 33.1/34.0 | 38.5/44.0 | **50.3/62.0** | 42.1/48.5 | 44.6/54.5 | **51.4/66.5** |
| | Enh | 24.4/46.0 | 10.0/26.0 | **25.6/46.0** | 5.5/4.0 | Fail | **7.8/4.0** | 20.1/44.0 | 24.3/46.0 | **24.3/46.0** | 7.1/24.0 | 8.9/22.0 | **8.9/28.0** | 14.3/29.5 | 10.8/22.5 | **16.7/31.0** |
| TN-enabled | Deg | **47.8/44.0** | 28.3/24.0 | **47.8/44.0** | 17.2/32.0 | 21.9/34.0 | **25.8/38.0** | 63.9/78.0 | 59.2/70.0 | **66.3/78.0** | 9.5/0.0 | 7.7/6.0 | **18.9/18.0** | 34.6/38.5 | 29.3/33.5 | **39.7/44.5** |
| | Enh | 13.3/34.0 | 15.0/34.0 | **17.2/36.0** | **6.3/4.0** | Fail | **6.3/4.0** | 10.7/24.0 | 24.3/44.0 | **24.3/44.0** | 5.3/12.0 | 5.3/14.0 | **5.3/14.0** | 8.9/18.5 | 10.0/21.5 | **13.3/24.5** |
| QRNCA-enabled | Deg | **48.9/46.0** | 31.1/28.0 | **48.9/46.0** | 9.4/16.0 | 18.8/30.0 | **21.1/32.0** | 60.4/70.0 | 42.6/46.0 | **65.0/78.0** | Fail | 6.5/6.0 | **6.5/6.0** | 30.3/31.5 | 24.8/27.5 | **35.4/40.5** |
| | Enh | 13.9/34.0 | 15.0/34.0 | **16.7/38.0** | 3.9/2.0 | Fail | **6.3/2.0** | 7.1/16.0 | 18.9/32.0 | **18.9/32.0** | 2.4/8.0 | 5.3/16.0 | **5.3/16.0** | 6.8/15.0 | 8.8/19.5 | **11.8/22.0** |

### LLaMA 2-13B

| Method | Eval | NER Good | NER Bad | NER Both | Chunking Good | Chunking Bad | Chunking Both | Sentiment Good | Sentiment Bad | Sentiment Both | Commonsense Good | Commonsense Bad | Commonsense Both | AVERAGE Good | AVERAGE Bad | AVERAGE Both |
|---|---|---|---|---|---|---|---|---|---|---|---|---|---|---|---|---|
| NeuronLLM | Deg | 29.9/29.3 | 28.1/26.7 | **32.6/33.3** | 29.6/40.0 | 32.6/40.0 | **28.8/46.7** | 6.4/3.6 | 22.1/20.0 | **36.6/41.8** | 25.3/28.8 | 21.2/25.8 | **33.8/37.9** | 22.8/25.4 | 26.0/28.1 | **33.0/39.9** |
| | Enh | **10.0/6.7** | 2.7/2.7 | **10.0/6.7** | **15.9/11.1** | 3.8/4.4 | **15.9/11.1** | 2.9/0.0 | 0.6/0.0 | **2.9/0.0** | | | | 7.9/4.8 | 3.7/3.3 | **9.2/7.1** |
| TN-enabled | Deg | Fail | 10.9/12.0 | **10.9/13.3** | 15.2/20.0 | 31.8/44.4 | **31.8/44.4** | Fail | 4.1/1.8 | **4.1/1.8** | 6.1/9.1 | 5.6/10.6 | **14.1/19.7** | 4.6/6.1 | 13.1/17.2 | **15.2/19.8** |
| | Enh | **7.2/6.7** | 3.2/1.3 | **7.2/6.7** | 12.1/15.6 | 15.2/8.9 | **16.7/20.0** | 5.2/3.6 | Fail | **3.5/7.3** | 2.0/1.5 | 3.5/3.0 | **4.0/3.0** | 6.6/6.9 | 5.5/2.4 | **7.9/9.3** |
| QRNCA-enabled | Deg | Fail | 10.9/12.0 | **11.8/14.7** | 12.1/11.1 | 31.8/44.4 | **31.8/44.4** | Fail | 2.9/3.6 | **2.9/3.6** | 5.1/9.1 | 6.1/10.6 | **11.1/19.7** | 4.0/4.3 | 12.9/17.7 | **14.4/20.6** |
| | Enh | **7.2/6.7** | 3.2/1.3 | **7.2/6.7** | 9.9/11.1 | 13.6/8.9 | **13.6/13.3** | 3.5/1.8 | Fail | **2.3/5.5** | 3.0/1.5 | 4.0/3.0 | **5.1/6.1** | 5.9/5.3 | 5.2/2.4 | **7.1/7.9** |

### Baichuan 2-7B

| Method | Eval | NER Good | NER Bad | NER Both | Chunking Good | Chunking Bad | Chunking Both | Sentiment Good | Sentiment Bad | Sentiment Both | Commonsense Good | Commonsense Bad | Commonsense Both | AVERAGE Good | AVERAGE Bad | AVERAGE Both |
|---|---|---|---|---|---|---|---|---|---|---|---|---|---|---|---|---|
| NeuronLLM | Deg | 34.0/40.3 | 24.4/27.8 | **63.6/73.6** | 47.2/59.6 | 23.3/31.6 | **50.3/64.9** | **46.0/51.7** | 39.8/55.2 | **46.0/51.7** | **56.7/74.6** | 27.9/28.4 | **56.7/74.6** | 46.0/56.6 | 28.9/35.8 | **54.2/66.2** |
| | Enh | 24.4/20.8 | 19.1/15.3 | **25.8/23.6** | 10.7/7.0 | 15.7/8.8 | **15.1/12.3** | 36.0/25.9 | 32.3/19.0 | **40.4/29.3** | 6.0/9.0 | **10.0/10.4** | **10.0/10.4** | 19.3/15.7 | 19.3/13.4 | **22.8/18.9** |
| TN-enabled | Deg | 7.2/9.7 | 10.0/12.5 | **22.0/23.6** | 47.2/59.6 | 44.7/56.1 | **48.4/59.6** | 3.7/1.7 | 8.7/15.5 | **32.9/44.8** | 7.0/6.0 | 7.5/7.5 | **30.3/32.8** | 16.3/19.3 | 17.7/22.9 | **33.4/40.2** |
| | Enh | 12.4/13.8 | 23.4/16.7 | **23.9/19.4** | 8.8/10.5 | **15.7/19.3** | **15.7/19.3** | 11.2/1.7 | **28.0/13.8** | **28.0/13.8** | 1.5/4.5 | 7.0/7.5 | **8.5/7.5** | 8.5/7.6 | 18.5/14.3 | **19.0/15.0** |
| QRNCA-enabled | Deg | 2.9/2.8 | 12.9/13.9 | **28.7/29.2** | 47.2/59.6 | 40.9/45.6 | **47.2/59.6** | 5.6/5.2 | 18.0/27.6 | **38.5/51.7** | 18.9/23.9 | 23.4/25.4 | **22.9/26.9** | 18.7/22.9 | 23.8/28.1 | **34.3/41.9** |
| | Enh | 12.4/12.5 | **23.9/18.1** | **23.9/18.1** | Fail | Fail | **8.8/7.0** | 9.3/1.7 | **19.3/6.9** | **19.3/6.9** | Fail | 6.0/4.5 | **5.5/7.5** | Fail | 12.2/6.5 | **14.4/9.9** |

## C.4 Sensitivity Analysis of the Intervention Budget

We evaluated the robustness of NeuronLLM to the intervention budget $K$. As demonstrated in Table 7, our method consistently outperforms competing state-of-the-art methods TN and QRNCA across all budget settings.

Table 7: Sensitivity analysis of the intervention budget $K$. Results show RAC/RCC for both enhancement (Enh) and degradation (Deg) performance across varying intervention budgets from 10 to 500 neurons (The task here is Commonsense Reasoning and the model is LLaMA 2-7B). NeuronLLM consistently outperforms competing SOTA methods TN and QRNCA across all budget settings, demonstrating its significant robustness to hyperparameter settings. Notably, NeuronLLM achieves with only 10 neurons the same control effectiveness that competing methods require 10× more neurons to attain (such superiority can also be observed in the results of NeuronLLM at budget 25 and 50 vs. TN/QRNCA at budget 250 and 500), demonstrating the effectiveness of NeuronLLM in identifying task-relevant neurons. For NeuronLLM, control effectiveness substantially improves from budget 10 to 100, then exhibits diminishing returns. In contrast, baseline methods show slower and more unstable improvement patterns, with occasional failed control.

| Intervention Budget $K$ | | | | | | | |
|---|---|---|---|---|---|---|---|
| | $K = 10$ | | $K = 25$ | | $K = 50$ | | $K = 100$ | |
| | Enh | Deg | Enh | Deg | Enh | Deg | Enh | Deg |
| NeuronLLM | **7.1/18.0** | **17.2/12.0** | **5.9/22.0** | **32.2/30.0** | **6.5/26.0** | **42.6/52.0** | **8.9/28.0** | **50.3/62.0** |
| TN | 1.2/4.0 | Fail | 0.6/6.0 | 2.4/0.0 | 4.7/10.0 | Fail | 5.3/12.0 | 9.5/0.0 |
| QRNCA | 1.2/6.0 | 0.0/0.0 | 0.6/6.0 | 2.4/0.0 | 3.0/6.0 | Fail | 2.4/8.0 | Fail |
| | $K = 150$ | | $K = 200$ | | $K = 250$ | | $K = 500$ | |
| | Enh | Deg | Enh | Deg | Enh | Deg | Enh | Deg |
| NeuronLLM | **11.2/28.0** | **52.1/64.0** | **11.2/28.0** | **52.7/66.0** | **10.7/26.0** | **52.1/64.0** | **10.1/24.0** | **52.7/66.0** |
| TN | 5.3/16.0 | 20.1/14.0 | 4.7/14.0 | 25.4/20.0 | 4.1/18.0 | 22.5/16.0 | 5.9/22.0 | 32.5/30.0 |
| QRNCA | 3.0/6.0 | Fail | 1.2/6.0 | 13.0/0.0 | 3.0/12.0 | 17.8/4.0 | 4.7/16.0 | 29.6/24.0 |

## C.5 Task-dependent neuron functionality

To show the task-dependent nature of neuron functionality which is discussed in Section 4.6, we conducted the following experiments on LLaMA 2-7B: For the Commonsense Reasoning task, we selected 100 good neurons and 400 bad neurons (task-specific), which were able to enhance the Commonsense Reasoning task with 28% RCC. For the SST (Sentiment) task, we selected 720 good neurons and 480 bad neurons, which similarly enhanced the Sentiment task with comparable RCC (30%). Then we check how the neurons identified for one task affect the other task.

The results in Table 8 shows the cross-task effects, demonstrating the task-dependent relationship where enhancing task-specific neurons of one task can negatively impact the performance of the other task, which means that neurons beneficial for one task may be detrimental to another, and vice versa. This further validates our functional antagonism hypothesis in LLMs.

Table 8: Cross-task effects between Sentiment Analysis and Commonsense Reasoning on LLaMA 2-7B. Values show RAC/RCC percentages when enhancing task-specific neurons identified from one task and evaluating performance on different tasks. Rows indicate the source task of intervened neurons, columns show the evaluated tasks.

| Intervened Neurons | Performance Change | |
|---|---|---|
| | Sentiment Task | Commonsense Task |
| Sentiment Neurons | 10.7%/30.0% | -11.8%/-6.0% |
| Commonsense Neurons | -18.3%/-18.0% | 10.7%/28.0% |

## C.6 STATISTICS OF TASK-RELEVANT NEURONS

We visualize the distribution of task-relevant neurons across different layers for LLaMA 2-7B, Baichuan 2-7B, and LLaMA 2-13B in Figures 5, 6, and 7.

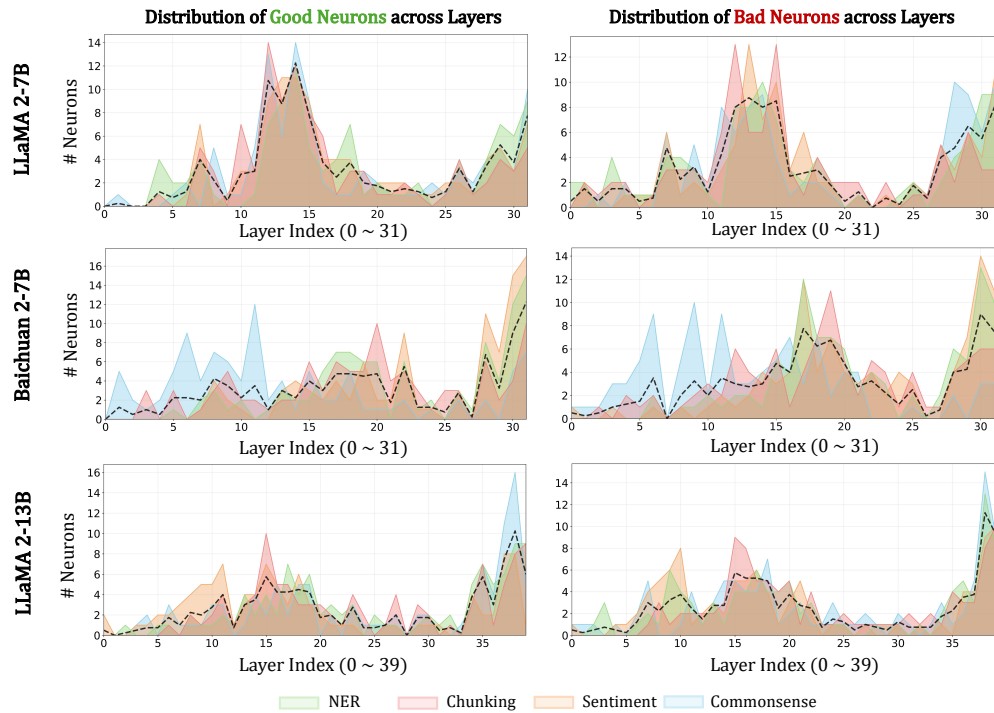

Figure 5: Distribution of the Top-100 "Good" (left) and Top-100 "Bad" (right) neurons identified by NeuronLLM across different tasks (plotted in different colors) and LLMs. The black dashed lines represent averaged distributions across four tasks. For Baichuan 2-7B, we find an interesting phenomenon that its task-relevant neurons of the Commonsense Reasoning task are more located in its earlier to middle layers compared to other tasks. Overall, we can clearly observe the concentration of good and bad neurons in middle and top layers, especially for LLaMA 2-7B (top row) and LLaMA 2-13B (bottom row), as indicated by the black dashed lines. Remarkably, good and bad neurons exhibit highly similar distribution patterns, suggesting they are functionally co-located in adjacent layers. Is is also worth noting that Baichuan 2-7B and LLaMA 2-13B exhibits slightly more task-relevant neurons in the later layers compared to middle layers, which may suggest their increased reliance on deeper processing stages. Taken together, these patterns align closely with previous findings (Li et al., 2025; Chen et al., 2025), indicating that the mechanisms related to task-execution mainly appear in the middle to later stages of LLMs.

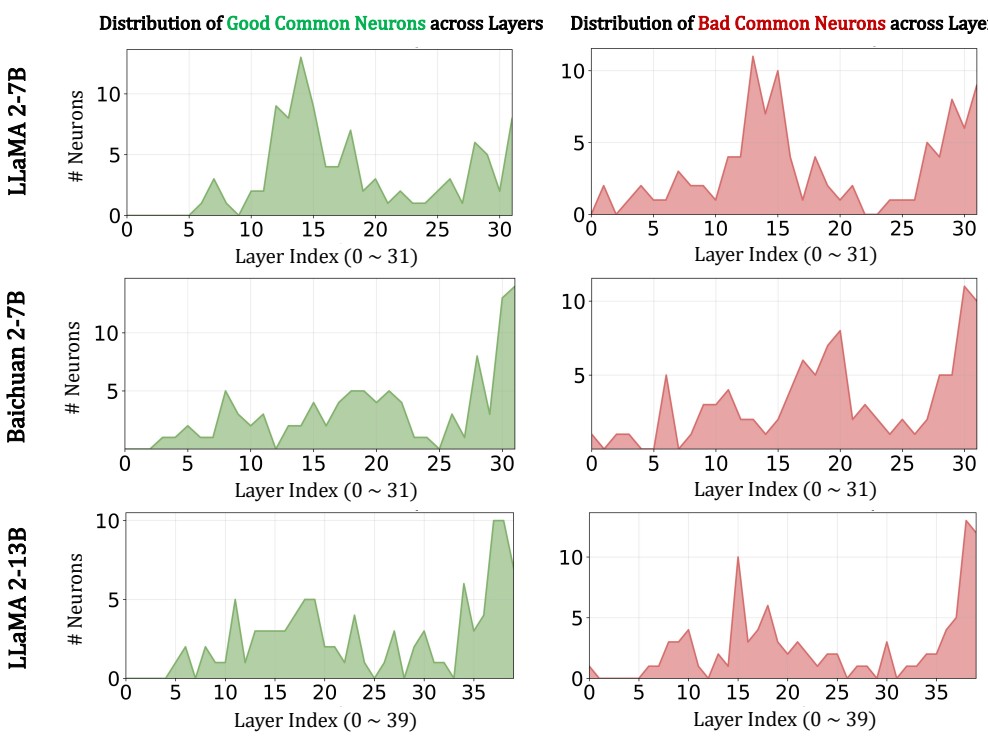

Figure 6: Distribution of the Top-100 common "Good" (left) and Top-100 common "Bad" (right) neurons identified by NeuronLLM across different model sizes. Similarly to the distribution of task-relevant neurons shown in Figure 5, these good and bad common neurons concentrate primarily in middle and top layers, as is evident from the larger colored areas in the middle/right parts in these plots.

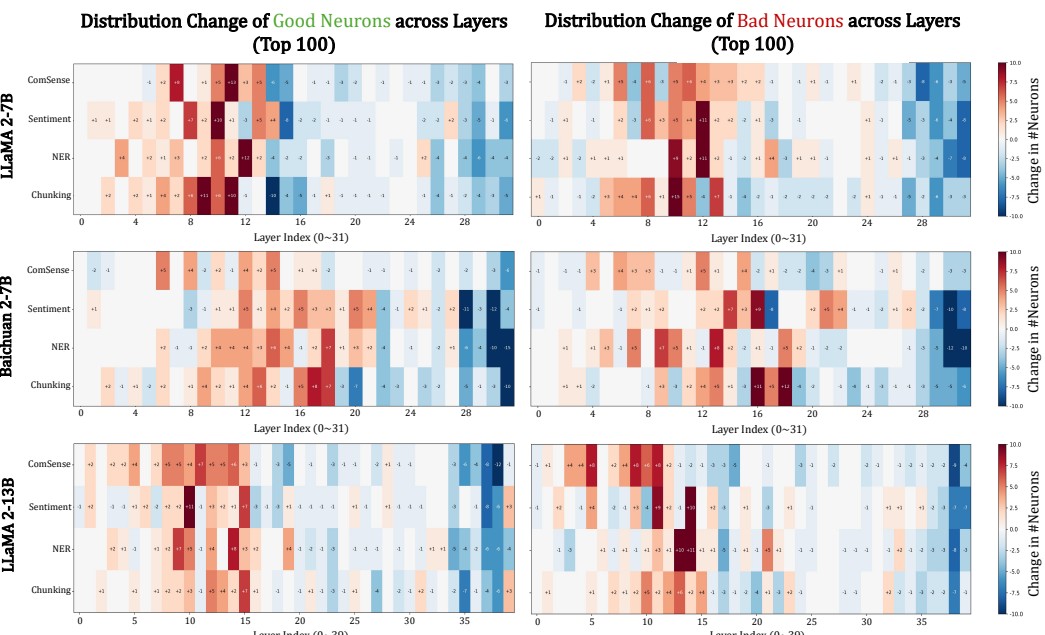

Figure 7: Layer distribution change of "Good" and "Bad" neurons after removing common neurons from the top-100 task-relevant neurons and refilling with subsequent task-specific neurons to maintain a total number of 100 neurons. The heatmaps show the difference between filtered and original distributions across layers, with positive values (red) indicating increased concentration and negative values (blue) indicating decreased concentration. Notably, the latter half of the model shows significant decreases in task-relevant neurons, revealing that many common neurons (either good or bad) that are crucial across different tasks reside in deeper layers. In contrast, the earlier layers exhibit increases in task-specific neurons, suggesting that the front layers might tend to encode task-specific mechanisms. Similar patterns are observed for both good and bad neurons across different tasks, model families and sizes.

## C.7 Enhancement vs. Degradation Asymmetry

Figure 8 provides an intuitive explanation of the enhancement vs. degradation asymmetry found in Section 4.5 (Figure 4). As the illustration shows, both Task X and Task Y require abilities C and D, but ability C is currently too weak to meet Task Y's threshold requirements and thus remains unutilized by Task Y, while Task X can still use it. When we excite Task X's task-specific neurons (strengthening abilities A, B, C), the enhanced ability C now surpasses Task Y's minimum threshold, causing Task Y's performance to suddenly improve as it begins utilizing this previously inaccessible ability. Conversely, degradation differs: silencing Task X's task-specific neurons (impairing abilities A, B, C) has minimal impact on Task Y since Task Y was not utilizing these abilities in the first place. This highlights that in complex systems like LLMs, enhancement and degradation are not simply inverse operations, considering the intricate interaction mechanisms between neurons. Through tools like NeuronLLM, we are able to explore LLM internals at the neuron level and observe such intriguing phenomena. We look forward to future research that can provide more theoretical rather than intuitive explanations for the underlying mechanisms between LLMs neurons of different roles, but this lies beyond the scope of this work.

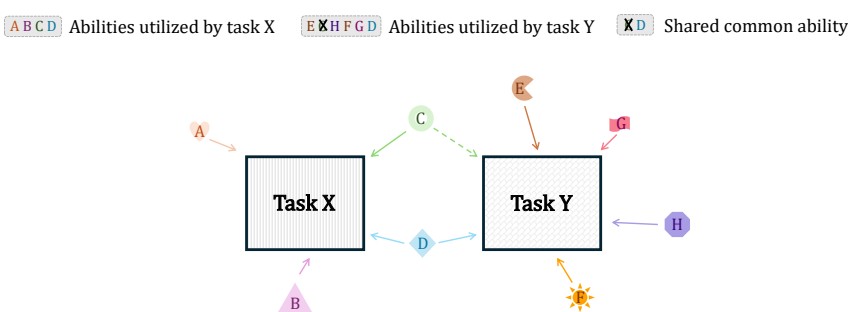

ability C is required for both task X and Y, but not utilized because it's currently too weak for task Y

Figure 8: Intuitive explanation for the enhancement vs. degradation asymmetry.

## C.8 COMPARISON OF COMPUTATIONAL EFFICIENCY

Figure 9 shows the comparison of computational efficiency between our Cross-Entropy-based Contrastive Neuron Scoring method in Eq. 3 (with QATT) and the summed log probabilities approach in Eq. 2 (without QATT). Through QATT transformation, the computation time remains stable at approximately 7 seconds per FF layer as it only requires backpropagation for a single token. In contrast, without QATT transformation, the computation cost grows linearly with the length of the answer sequence, as it requires computing the summed log probabilities for the complete output sequence. This demonstrates that QATT not only enables a complete view of response signals from LLMs (providing both correct and incorrect answers) and unifies the output format, but also significantly improves computational efficiency in practice.

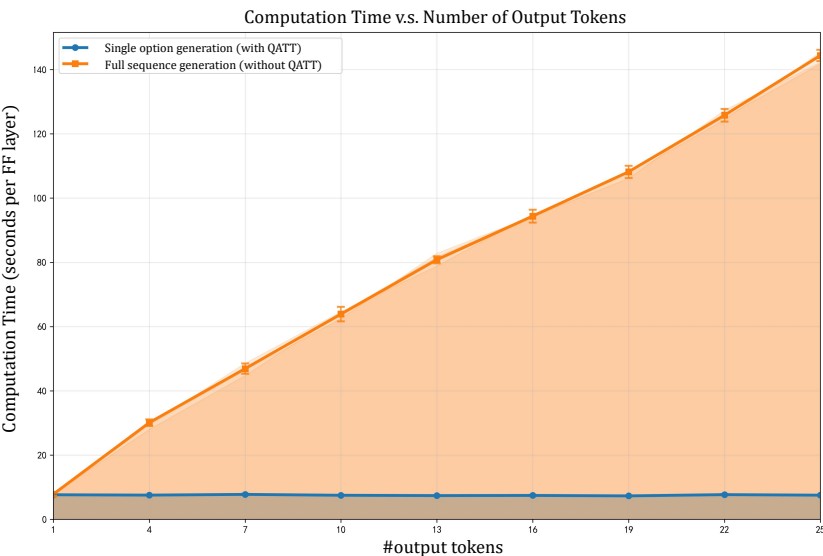

Figure 9: Comparison of computational efficiency of our neuron scoring method against existing methods. The blue line represents single option generation of our method (with QATT transformation), while the orange line represents the alternative method in existing methods that requires full sequence generation (without QATT transformation). Each line shows the average computation time across three runs.

## C.9 DISTRACTOR GENERATION DETAILS

Here we provide detailed descriptions of how distractors are generated for each of the four tasks:

1. **Named Entity Recognition (NER)**: We randomly sample three different entities from other possible entity types as distractors. For example, if the correct answer is a person entity, the distractors might include organization, location, and miscellaneous entities.

2. **Chunking**: We utilize advanced LLMs (specifically Gemini 2.5 Pro) to generate distractors using carefully designed prompts. The prompt is structured as: "Based on the correct chunking segmentation, generate three additional incorrect chunking options." The generated distractors are then manually reviewed to ensure the quality of the questions.

3. **Sentiment Classification**: The total possible answers are fixed to four categories: positive, negative, neutral, and not sure. No additional distractor generation is required.

4. **Commonsense Reasoning**: Since the original dataset already follows a multiple-choice format, we directly use the existing answer choices provided in the dataset.

## C.10 VISUAL EXAMPLES FOR OPTION PROBABILITIES CHANGE CAUSED BY MODEL CONTROL

As shown in Figure 10, for both enhancement and degradation, our method can more effectively control the probability gap between correct and wrong options in the desired directions.

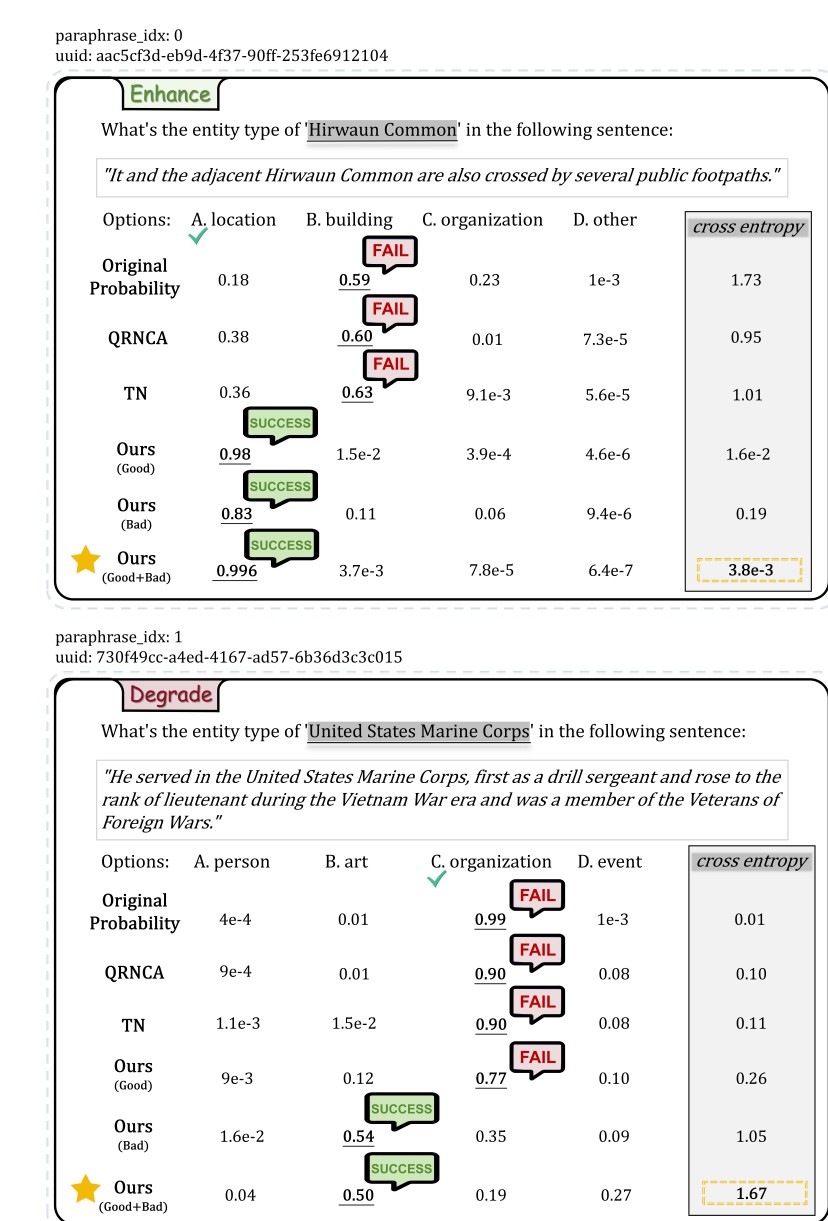

Figure 10: Visual examples of option probabilities before control (row "Original Probability") the original model with QRNCA-detected neurons and after control (row "QRNCA"), TN-detected neurons (row "TN"), NeuronLLM-detected good neurons (row "Ours (Good)"), bad neurons (row "Ours (Bad)"), and both kinds of neurons (row "Ours (Good+Bad)"). The probabilities are after softmax normalization over the four options. Underlined value indicates the highest probability in each row. For enhancement, if the highest probability option is the correct answer which is marked by ✓, the control is successful. For degradation, if the highest probability option is a wrong answer, the control is successful; Our method (good+bad) achieves the best control performance in both scenarios, with the lowest/highest cross-entropy between model prediction and true label for enhancement/degradation compared to other methods.

## D    THE USE OF LARGE LANGUAGE MODELS

Large Language Models (LLMs) were utilized in two main capacities during this research. Firstly, we employed LLMs as an auxiliary tool for grammatical correction and to improve the overall readability of the manuscript. Secondly, LLMs played a role in the data creation process for the simplified chunking task. Specifically, they were used to generate distractor answer choices, which helped in constructing a dataset suitable for our experimental needs as described in Section A.1.

