# OpenReview forum: "Identifying Good and Bad Neurons for Task-Level Controllable LLMs"
_ICLR.cc/2026/Conference — ICLR 2026 Conference Withdrawn Submission_

### Official Review · Reviewer_GAoB · 2025-10-27

**Soundness:** 1
**Presentation:** 2
**Contribution:** 2
**Rating:** 4
**Confidence:** 4

**Summary:**

This paper introduces NeuronLLM, a framework designed to improve task-level understanding and control of large language models (LLMs) by identifying neurons with supportive ("good") and inhibitory ("bad") roles for downstream NLP tasks. NeuronLLM employs a Question-Answering-based Task Transformation (QATT) to unify diverse task formats into a multiple-choice QA protocol, and a Contrastive Neuron Identification (CNI) module leveraging a cross-entropy-based additive scoring approach to attribute neuron importance. Extensive results are reported for three LLMs (LLaMA 2-7B, Baichuan 2-7B, LLaMA 2-13B) across four NLP tasks, with control-based interventions used to validate neuron function. The paper emphasizes the biological analogy of "functional antagonism" and claims superior empirical performance over recent SOTA neuron attribution methods.

**Strengths:**

1.	Good writing and motivation illustration: This paper highlights the reason why we need for not only activating the supporting neurons but also removing the neurons which declines the performance from both biological and mathematical view.

**Weaknesses:**

1.	Lack of innovation: The main method of the paper is based on a biological concept and existing methods for enhancing and reducing activation values, but more often than not, biological concepts are applied to the shell of existing theories, lacking unique innovation.
2.	The comparison of baselines lacks fairness: The main models you compared are Llama2-7B, Llama2-13B, and Baichuan2-7B. Why not use some newer and better open-source models, such as Qwen, Llama3, and other series of models. And some performance indicators have increased from 7% of the baseline to 63%, but the authenticity of the results needs to be measured.
3.	The representativeness of dataset selection needs to be considered: you only selected one dataset for each task, how can you ensure that the selected dataset represents that task?

**Questions:**

See weaknesses.

---

> ### Author Response · Authors · 2025-11-22
> **response to comments #1~3**
>
> We sincerely appreciate your constructive comments. Please see our response to your comments one by one below.
>
> >**Comment #1:** Biological concepts are applied to the shell of existing theories, lacking unique innovation.
>
> We thank the reviewer for the comments. We would like to clarify that while our work is inspired by biological concepts of functional antagonism, our contributions go beyond merely applying these concepts to existing theories. Specifically, we introduce a new contrastive neuron attribution method that redefines the attribution objective using cross-entropy, effectively mitigating the "collateral effect" problem observed in prior works like QRNCA (we kindly refer you to our response to ***Reviewer qwJC Comment #1***). This methodological and conceptual innovation revealed and addressed a key overlooked issue in previous integrated gradients based neuron attribution methods for LLMs. Also, we believe the functional antagonism perspective itself is not a trivial finding, as it provides an interesting lens to understand the internal mechanisms of neurons in LLMs, which has not been studied before. Notably, our experiments in the main paper show that the "Both" strategy which simultaneously controls both good and bad neurons consistently outperforms only controlling good or bad neurons, even when the control budget is the same. This finding is valuable for future works on neuron identification and control, which suggests that considering the functional antagonism is important for neuron-level understanding of LLMs. We hope this clarification addresses your concerns and we'll appreciate if you could reconsider your rating.
>
> >**Comment #2:** Choice of Baseline Models
>
> We appreciate the reviewer's suggestion regarding the choice of baseline models. We selected Llama 2-7B, Llama 2-13B, and Baichuan 2-7B as they are widely used and well-established open-source models, providing a solid foundation for evaluating and replicating our method. In addition, both our method and the competing baselines are built upon the same LLM in each comparison to ensure fairness in all our experiments.
>
> Large performance variations are expected due to the change in the datasets and the inherent complexity of the problem. We have carefully and thoroughly cross-checked our implementation and experiment results, through which we are 100\% confident about the authenticity of the reported results. We have also provided our codes and data in our submission for the reviewers' verification if needed.
>
> >**Comment #3:** Dataset Selection.
>
> We thank the reviewer for raising this question. As illustrated in Section 4.1 and Appendix A.1 of the main paper, we selected very popular and widely-used datasets for each task to provide a fair evaluation of our method across diverse task types. Due to the computational cost of neuron attribution and evaluation, we have to do a tradeoff between the number of datasets and the diversity of task types, which we choose to prioritize the latter to better validate the general applicability of our method across different tasks. We hope this clarifies our dataset selection rationale.

---

### Official Review · Reviewer_51Qj · 2025-10-29

**Soundness:** 2
**Presentation:** 2
**Contribution:** 3
**Rating:** 6
**Confidence:** 4

**Summary:**

The paper addresses the problem that the influence of neurons on task execution in large language models (LLMs) is opaque, and existing methods focus only on supportive neurons while lacking adaptability to multi-task scenarios. It proposes the NeuronLLM framework, which draws on the biological principle of functional antagonism, viewing task performance as the combined effect of “good” neurons that facilitate and “bad” neurons that inhibit task completion. The framework employs the Question-Answering-based Task Transformation (QATT) module to unify various tasks into a question-answering format, and the Contrastive Neuron Identification (CNI) module to identify “good” and “bad” neurons using a cross-entropy-based contrastive scoring method. Experiments show that NeuronLLM outperforms existing methods across multiple LLMs and four NLP tasks, providing a more comprehensive identification of task-relevant neurons and revealing the functional organization of the models.

**Strengths:**

1.The paper proposes the NeuronLLM framework, which leverages two opposing types of neurons: task-supporting “good” neurons and task-inhibiting “bad” neurons, to achieve overall task-level control of LLMs.
2.The paper introduces a Question-Answering-based Task Transformation module that unifies various tasks into a question-answering format, enabling NeuronLLM to interpret LLMs under different tasks.
3.The paper presents a contrastive neuron identification module, which uses a novel cross-entropy-based contrastive scoring method to identify “good” and “bad” neurons, providing a holistic perspective on neuron analysis.
4.The paper conducts comprehensive experiments across LLMs of different scales and families, demonstrating that NeuronLLM significantly outperforms existing methods in identifying task-relevant neurons.

**Weaknesses:**

1.The paper uses performance change metrics RAC and RCC in the comparative experiments, which do not directly show the effects of Degrade and Enhance, i.e., whether the model’s performance decreases or improves. It is recommended to add metrics that can visually indicate performance increases and decreases.
2.The paper proposes a Question-Answering-based Task Transformation module, which unifies diverse tasks into a QA format to ensure that the neuron attribution method can model a consistent output objective. However, the ablation study does not validate the effectiveness of this module. It is recommended to add relevant experiments to evaluate the impact of this module on the overall framework.
3.In the ablation study regarding joint modeling of good and bad neurons, it is not specified whether the control of “Good” or “Bad” neurons is Exciting or Silencing. Similarly, in the “Both” strategy, it is unclear whether it acts as an enhancer or a degrader.
4.Equation (1) in the paper explains that the contribution of a neuron to the objective function is approximated via integrated gradients, but the parameter kkk is not explained. It is recommended to provide an explanation to make it easier for readers to understand.
5.Figure 1 shows the framework of the proposed NeuronLLM method, including the QA-based Task Transformation module, the Contrastive Neuron Identification module, and the Neuron Intervention and Evaluation module. However, the methods of the Contrastive Neuron Identification module and the Neuron Intervention and Evaluation module are not illustrated in the figure. Additionally, the font size of the QA-based Task Transformation module is too small; increasing the font size is recommended.
6.The paper does not provide the code for the NeuronLLM framework, resulting in low reproducibility. It is recommended that the authors release the implementation for verification and wider use.

**Questions:**

1.The paper proposes a Question-Answering-based Task Transformation module, which aims to unify diverse tasks into a QA format to ensure that the neuron attribution method can model a consistent output objective. How do the authors validate the effectiveness of this module?
2.Table 2 shows the results of integrating existing neuron scoring methods into the NeuronLLM framework. In this experiment, the authors replaced the proposed ACE scoring method with the TN/QRNCA scoring methods. Can the TN/QRNCA scoring methods distinguish between good and bad neurons?
3.Does the NeuronLLM framework proposed in the paper, which models 'good' and 'bad' neurons, include any visualization results? Has it attempted to fine-tune downstream tasks based on the identified 'good' and 'bad' neurons (such as using an enhancer strategy)? If so, what are the specific fine-tuning methods used and what are the resulting effects?

---

> ### Author Response · Authors · 2025-11-22
> **response to comment #1**
>
> We sincerely appreciate your constructive and positive comments. Please see our response to your comments one by one below.
>
> >**Comment #1:** More Straightforward Illustration of Model Editing.
>
> We appreciate this helpful feedback. To better illustrate and compare how editing the neurons detected by different methods affects model's prediction. We have added Figure 10 in the revised pdf which compares the effects of editing neurons detected by different methods. For your convenience, we also provide two simplified tables below containing the key content from Figure 10. These tables display option probabilities for the original model (row "Original Probability") and after editing neurons identified by QRNCA (row "QRNCA"), TN (row "TN"), and our method. For our method, we show the effects of editing good neurons ("Ours (Good)"), bad neurons ("Ours (Bad)"), and both ("Ours (Good+Bad)"). The probabilities are after softmax normalization over the four options. Bold value indicates the highest probability in each row. For enhancement, if the highest probability option is the correct answer, the control is successful. For degradation, if the highest probability option is a wrong answer, the control is successful; Our method (good+bad) achieves the best control performance in both scenarios, with the lowest/highest cross-entropy between model prediction and true label for enhancement/degradation compared to other methods.
>
> **Table 2: Enhance Example:**
> *Question: What's the entity type of 'Hirwaun Common' in the following sentence: "It and the adjacent Hirwaun Common are also crossed by several public footpaths."*
>
> | Method | A. location (Correct) | B. building | C. organization | D. other | Cross Entropy |
> | :--- | :--- | :--- | :--- | :--- | :--- |
> | Original Probability | 0.18 | **0.59** (FAIL) | 0.23 | 1e-3 | 1.73 |
> | QRNCA | 0.38 | **0.60** (FAIL) | 0.01 | 7.3e-5 | 0.95 |
> | TN | 0.36 | **0.63** (FAIL) | 9.1e-3 | 5.6e-5 | 1.01 |
> | Ours (Good) | **0.98** (SUCCESS) | 1.5e-2 | 3.9e-4 | 4.6e-6 | 1.6e-2 |
> | Ours (Bad) | **0.83** (SUCCESS) | 0.11 | 0.06 | 9.4e-6 | 0.19 |
> | Ours (Good+Bad) | **0.996** (SUCCESS) | 3.7e-3 | 7.8e-5 | 6.4e-7 | **3.8e-3** (the lowest) |
>
> **Table 3: Degrade Example:**
> *Question: What's the entity type of 'United States Marine Corps' in the following sentence: "He served in the United States Marine Corps, first as a drill sergeant and rose to the rank of lieutenant during the Vietnam War era and was a member of the Veterans of Foreign Wars."*
>
> | Method | A. person | B. art | C. organization (Correct) | D. event | Cross Entropy |
> | :--- | :--- | :--- | :--- | :--- | :--- |
> | Original Probability | 4e-4 | 0.01 | **0.99** (FAIL) | 1e-3 | 0.01 |
> | QRNCA | 9e-4 | 0.01 | **0.90** (FAIL) | 0.08 | 0.10 |
> | TN | 1.1e-3 | 1.5e-2 | **0.90** (FAIL) | 0.08 | 0.11 |
> | Ours (Good) | 9e-3 | 0.12 | **0.77** (FAIL) | 0.10 | 0.26 |
> | Ours (Bad) | 1.6e-2 | **0.54** (SUCCESS) | 0.35 | 0.09 | 1.05 |
> | Ours (Good+Bad) | 0.04 | **0.50** (SUCCESS) | 0.19 | 0.27 | **1.67** (the highest) |
>
> As shown for both enhancement and degradation, our method can more effectively control the probability gap between correct and wrong options in the desired directions. Especially, as explained in our response to ***Reviewer qwJC Comment #1*** (Table 1), we reveal the collateral effect of QRNCA which is one main reason for why it performs worse. We will include these new results in the revised paper.

---

> > ### Author Response · Authors · 2025-11-22
> > **response to comment #2**
> >
> > >**Comment #2:** Evaluate the Impact of QATT Module.
> >
> > We appreciate this valuable suggestion. As explained in the main paper, the QATT module essentially brings two advantages: (1) It provides a richer context including contrastive correct and wrong answers which better illicit the model's knowledge about the question, and an easy way to create proxy questions (shuffling the options), avoiding model guessing the answer by chance. Hence we can use the more precise cross-entropy-based neuron attribution. (2) We can avoid attributing to the whole answer sequence, which can be multiple tokens long, by focusing only on the option token.
> >
> > To further dissect and validate the contribution of these two advantages, we conduct additional ablation experiments:
> >
> > (1) We remove the multi-choice format by directly asking the question without providing optional answers (hence also no proxy questions) and attribute the integrated gradients to the answer (as shown in the formula (2) of the main paper). To focus on the influence of the contrastive answers, here we keep only those questions whose answers can be reduced to one token. Specifically, we do experiments on NER and Commonsense tasks. The results (as in Table 4) show that, without the contrastive information provided by the options and the proxy questions, the control performance gets significantly worse, highlighting their importance in our method.
> >
> > **Table 4:** Ablation study on the impact of multi-choice format with contrastive answers and proxy questions. We compare plain QA format (without options & proxy questions), QRNCA, and our method. The control strategy here is "Both". Results show RAC/RCC metrics for degrade and enhance operations on NER and Commonsense tasks.
> >
> > | Task | Operation | Plain QA | QRNCA | NeuronLLM |
> > | :--- | :--- | :--- | :--- | :--- |
> > | NER | degrade | 20.1/19.6 | 48.9/46.0 | 53.3/64 |
> > | | enhance | 7.8/20.0 | 16.7/38.0 | 25.6/46.0 |
> > | Commonsense | degrade | 2.4/4.0 | 6.5/6.0 | 50.3/62.0 |
> > | | enhance | 1.2/2.0 | 5.3/16.0 | 8.9/28.0 |
> >
> > (2) We compare the control performance between single-token attribution and multi-token (sequence) attribution. Here we only use the Commonsense task because its answers are relatively easy to reduce or extend length.
> >
> > The results in Table 5 show that multiple-token attribution is totally not working. This is actually consistent with our analysis regarding the collateral effect in QRNCA. When attributing to multiple tokens, the answer space becomes exponentially larger (e.g. if the vocabulary size is $V$ and answer length is $L$, one specific answer is only one point in the large answer space of size $V^L$.), obviously making it very likely that neurons identified as good for increasing the probability of a specific answer sequence probably also increase the probabilities of many other irrelevant or even contradictory outputs in the massive answer space, leading to ineffective identification and control. Our QATT module avoids this problem by transforming the multi-token attribution into single-token attribution, fundamentally mitigating the collateral effect problem that would even get more severe as answer gets longer. We hope these additional analyses further validate the importance and effectiveness of our QATT module. We would also like to admit that, despite the advantages of QATT module demonstrated above, this transformation also brings burden on the context length, so for those tasks with very long answers, our QATT module may not be applicable. We will clarify this limitation in the revised paper.
> >
> > **Table 5:** Comparison of single-token vs. sequence attribution. Results show that multi-token (sequence) attribution completely fails (0/0), suggesting that focusing on single tokens attribution is crucial for effective neuron identification for LLM.
> >
> > | Operation | Single token | Sequence |
> > | :--- | :--- | :--- |
> > | degrade | 2.4/4.0 | 0/0 |
> > | enhance | 1.2/2.0 | 0/0 |

---

> > > ### Author Response · Authors · 2025-11-22
> > > **response to comments #3~8**
> > >
> > > >**Comment #3:** Specification the Control Method in Ablation Study.
> > >
> > > We apologize for the lack of clarity about presenting this part. For enhancement control, "Good" neurons are excited while "Bad" neurons are silenced. For degradation control, "Good" neurons are silenced while "Bad" neurons are excited. In ablation study, if there is only "Good" or "Bad" neurons (e.g. the "Good" and "Bad" columns in Table 2 and Table 3), the other set of neurons are simply not edited. We will clarify this in the revised paper.
> > >
> > > >**Comment #4:** Explaination for parameter K.
> > >
> > > Thanks for the comment. Integrated gradients are defined as the path integral of the gradients along the straightline path from the baseline activation 0 to the neuron's actual activation value. In practice this integral is approximated via summing the gradients at points occurring at
> > > sufficiently small intervals along the path. Therefore, $k$ refers to the $k_{th}$ sampled point. We will revise the paper to clarify this.
> > >
> > > >**Comment #5:** Improve the Visualization of Figure 1.
> > >
> > > We will improve the visual illustration for each module in Figure 1 and adjust the font-size for better visibility.
> > >
> > > >**Comment #6:** Code for Replication.
> > >
> > > The core code has been provided in the supplementary materials. In addition, we will release the full code upon paper acceptance for replication.
> > >
> > > >**Comment #7:** Can the TN/QRNCA scoring methods distinguish between good and bad neurons.
> > >
> > > Technically yes but they all ignore that, which is an overlooked point that we want to emphasize by this work. Moreover, as explained in our response to ***Reviewer qwJC Comment #1***, due to the "collateral effect" problem, their identification of good and bad neurons is less effective than ours. As evidenced by the experiments in our main paper, even when enabled with NeuronLLM, controlling the neurons detected by TN/QRNCA results in worse control performance. This highlights that besides controlling both good and bad neurons which is a more holistic edit strategy, the effectiveness of the identification method itself is also crucial.
> > >
> > > >**Comment #8:** Attemptations to Fine-tune Downstream Tasks Based on Identified Neurons.
> > >
> > > We agree that leveraging identified good and bad neurons for further downstream fine-tuning is an interesting direction, and we appreciate the reviewer's suggestion. However, in this work, we primarily focus on neuron identification, especially revealing the functional antagonism in LLMs at the neuron level. That's also why we use only simple interventions like silencing and exciting to facilitate a straightforward evaluation of the identified neurons. We suggest that fine-tuning based on good and bad neurons could be equipped with more intricate neuron steering operation (e.g. weighted control instead of simply multiplying by 0 or 2), which we will explore in future work.

---

> > > > ### Comment · Reviewer_51Qj · 2025-11-26
> > > >
> > > > I thank the authors for their response. However, the key weaknesses of the manuscript persist. Given that no downstream task experiments were conducted and the proposed method demonstrates relatively limited novelty, I will maintain my original rating.

---

### Official Review · Reviewer_JeJh · 2025-10-30

**Soundness:** 3
**Presentation:** 3
**Contribution:** 2
**Rating:** 4
**Confidence:** 4

**Summary:**

This paper introduces **NeuronLLM**, a novel framework for identifying task-relevant “good” and “bad” neurons in LLMs, inspired by the biological principle of *functional antagonism*. The authors argue that task performance is jointly determined by both supportive and inhibitory neurons, a perspective largely overlooked by prior work. NeuronLLM consists of two main components: (1) **QATT**, which transforms diverse tasks into a unified multiple-choice QA format, and (2) **CNI**, which employs a contrastive cross-entropy-based scoring method (ACE) to identify neurons with opposing roles. Extensive experiments across multiple LLMs and NLP tasks demonstrate that NeuronLLM significantly outperforms existing neuron attribution methods in both degrading and enhancing task performance through targeted neuron interventions.

**Strengths:**

- *Novel Conceptual Insight*: The introduction of “bad” (inhibitory) neurons alongside “good” ones, inspired by functional antagonism, provides a more holistic and biologically plausible view of LLM internal mechanisms.
- *Practical Framework Design**: The two-stage design (QATT + CNI) is well-motivated. QATT effectively standardizes diverse tasks, enabling consistent neuron analysis, while the ACE scoring in CNI naturally fits the QA format and captures both positive and negative contributions.
- *Comprehensive Evaluation*: The paper provides thorough experiments across multiple model families (LLaMA 2, Baichuan) and sizes (7B, 13B), four distinct NLP tasks, and several strong baselines. The results convincingly demonstrate NeuronLLM’s superiority.
- *Ablation and Analysis*: The paper includes valuable ablation studies (individual vs. joint intervention of good/bad neurons) and insightful analyses (common vs. task-specific neurons, enhancement vs. degradation asymmetry), which deepen the understanding of the framework and the phenomena it uncovers.

**Weaknesses:**

- *Limited Scope of Tasks*: While the framework is proposed for task-level understanding, the evaluated tasks are all classification or multi-choice QA after transformation. It remains unclear how well NeuronLLM generalizes to true *generation* tasks where the output space is open-ended and the “distractor” choices in QATT are not naturally defined. Neuron-Level Knowledge Attribution in Large Language Models (Yu et al., 2024) claims that the active neurons are related to task domains and its form, I wonder does the identified neurons changed by the variant domains or forms?
- *Justification of the “Bad” Neuron Concept*: The paper shows that silencing certain identified neurons degrades performance, which is a post-hoc validation. However, a more intrinsic characterization or explanation of *what these “bad” neurons are doing* is lacking. The concept is currently defined primarily by the intervention effect, lacking mechanism interpretation. In formula 3, integrated gradients are calculated on the gradient
$$\frac{\partial F}{\partial w_i^l} = \frac{\partial P(c^{*})}{\partial w_i^l}$$ However, the "good or bad" rating of a neuron does not solely depend on how it affects the score of the correct option $s_c^{\*}$, but on the difference between the gradient of $s_c^{\*}$ and the weighted average of the gradients of all option scores.
- *Computational Cost Discussion*: Although Figure 9 shows QATT improves efficiency compared to full-sequence generation, the overall cost of NeuronLLM (involving integrated gradients over multiple proxy questions and aggregation) is still non-trivial, especially for very large models. A discussion of the scalability and potential bottlenecks would be beneficial.
- *Ablation on QATT's Role*: The paper ablated the effect of good/bad neurons but did not thoroughly ablate the contribution of QATT itself. For instance, how crucial is the multi-choice format with distractors compared to a simpler QA format?

**Questions:**

- *Generalization to Generation Tasks*: How would NeuronLLM handle free-form generation tasks? Would the distractor choices in QATT become arbitrary or hard to define, and how might this impact the contrastive scoring in CNI?
 - *Circuit Related to Identified Neurons*: How to determine whether silencing certain identified neurons degrade by destroying circuits in the model or by identifying neurons? The paper assumes a direct causal relationship between good/bad neurons and their functions. However, in the distributed representation of neural networks, there is a high degree of co adaptation and redundancy between neurons. How to rule out the possibility that the identified "good" neurons are actually only highly correlated with the unknown neuron X that truly plays a causal role?
- *Intrinsic Nature of “Bad” Neurons*: Beyond their inhibitory effect on the target task, is there any intrinsic properties or patterns in the “bad” neurons? What’s the mechanism of good/bad neurons in LLMs?
- *Sensitivity to Distractor Quality*: The quality of distractor choices seems crucial for the contrastive scoring. How sensitive is NeuronLLM to the method of distractor generation (e.g., random sampling vs. LLM-generated)? Is there any correlation between distractor quality and neuron identification accuracy?
- *Comparison Broader Baselines*: The paper compares against recent neuron attribution methods (TN, QRNCA). Have you considered comparing with more general feature attribution methods (e.g., using integrated gradients directly on the original task format) to better isolate the gain from the good/bad neuron paradigm versus the QATT+ACE pipeline?

---

> ### Author Response · Authors · 2025-11-22
> **response to comments #1~4**
>
> We sincerely appreciate your constructive comments. Please see our response to your comments one by one below.
>
> >**Comment #1:** Ablation on QATT's Role.
>
> We have conducted additional ablation experiments to dissect and validate the contribution of the QATT module. Since this point was also raised by other reviewers, we have provided the detailed experiment results (Table 4 and Table 5) and analysis in our response to ***Reviewer 51Qj Comment #2***. To avoid redundancy, we kindly refer you to that section. The results demonstrate that both the contrastive answers and proxy questions provided by QATT are crucial for effective neuron identification and control, and focusing on single-token attribution is very important to avoid intensifying the "collateral effect" (please refer to our response to ***Reviewer qwJC Comment #1*** (Table 1) for detailed explanation). We hope these additional analyses further validate the importance and effectiveness of our QATT module. We will include these new results in the revised paper.
>
> >**Comment #2:** Concerns about Long-answer Generation Tasks
>
> We appreciate this valuable comment. Indeed, generation tasks include diverse scenarios. Although in theory we can transform them into multi-choice format by pairing correct answers with contrastive wrong answers, in practice, for tasks with really long answers (e.g., story generation), such transformation is more challenging. The reason is two-fold: (1) As the reviewer mentioned, designing "distractor" choices for long answers can be hard. (2) The context length gets much increased, which might exceed the model's context window limit.
>
> We agree with the reviewer that directly applying QATT to such tasks may not be feasible in practice and will clarify this limitation in the revised paper. Neuron identification methods for long-answer generation tasks needs to be specially designed and we hope the contrastive neuron identification idea and functional antagonism of LLM proposed in this work can still provide useful insights for future research.
>
> >**Comment #3:** Concerns about Task Format: Open-ended Generation v.s. Multi-choice QA
>
> We thank the reviewer for raising this important point. Our additional ablation experiments shown in Table 4 and Table 5 in our response to ***Reviewer 51Qj Comment #2*** validate that the task format indeed matters a lot for neuron attribution in LLMs. Specifically, the multi-choice QA format is much more effective than open-ended generation format for integrated-gradients-based neuron identification and control. The later suffers from the severe "collateral effect" problem (which is illustrated in our response to  ***Reviewer qwJC Comment #1***) for directly applying integrated gradients on the answer sequence, making it unable to accurately identify task-relevant neurons. In contrast, the multi-choice QA format brings two key advantages: (1) It allows us to avoid attributing to the whole answer sequence, which can be multiple tokens long, by focusing only on the option token, fundamentally mitigates the collateral effect problem which will gets more severe as answer gets longer. (2) It provides contrastive information from correct and wrong answers which better illicit the model's knowledge about the question, enabling more precise neuron attribution using cross-entropy-based scoring.
>
> >**Comment #4:** Computational Cost Discussion.
>
> We agree with the reviewer that the computational cost of NeuronLLM is an important consideration, especially when scaling to very large models. The computational cost is mainly due to the integrated gradient calculation which requires multiple forward and backward passes through the model for each proxy question. With one H200 GPU, in average, calculating neuron scores for one question (consisting of 3 proxy questions) on LLaMA 2-7B takes about 670 seconds. Note that this can actually be improved a lot from the engineering perspective, e.g. parallelizing the calculation across multiple GPUs, since the calculation for different proxy questions are independent. Considering our main focus in this work is to reveal the functional antagonism in LLMs at the neuron level and show a more precise contrastive neuron identification method, we didn't further optimize the computational efficiency, but will include the discussion in the revised paper.

---

> > ### Author Response · Authors · 2025-11-22
> > **response to comments #5~8**
> >
> > >**Comment #5:** Questions about Formula (3).
> >
> > Thanks for raising this important question. This is exactly the motivation of our cross-entropy-based scoring. In formula (3), the probability of the correct option is softmax-normalized over all four options hence the impact of a neuron on the probabilities of both correct and wrong answers are considered. As we wrote in line 269-271 of the main paper, this target function is mathematically equivalent to decreasing the cross-entropy between the model's predicted distribution over four options and the true label distribution. We will clarify this in the revised paper.
> >
> > >**Comment #6:** More Intrinsic Characterization of Bad Neurons.
> >
> > We appreciate this insightful comment. Due to the black-box nature of LLMs, fully understanding the intrinsic, mechanistic properties of bad neurons is very challenging. Hence we utilize classic intervention approaches to validate their oppose functions compared to good neurons in a post-hoc way. Our intuition is that bad neurons might encode misleading, task-unrelated features that distract the model from producing correct answers. These features might be beneficial for some tasks but not the evaluated one, since LLMs are trained to be a all-round player on diverse tasks with super large-scale corpora. This is evidenced by our experiments in line 471-474 and Table 8 of the main paper. We agree with the reviewer that more intrinsic characterization of bad neurons is definitely an interesting direction and we look forward to future works exploring this, e.g., leveraging  mechanistic interpretability methods to further analyze the good and bad neurons.
> >
> > >**Comment #7:** Circuit Related to Identified Neurons.
> >
> > We thank the reviewer for raising this important point. Indeed, due to the distributed representation and redundancy in neural networks, fully ruling out the possibility of co-adaptation and correlation is very challenging. To our best knowledge, there is currently no perfect method to completely disentangle correlation and causation in neuron-level analysis of LLMs. Based on our intervention experiments, our advice is carefully designing the attribution target can help mitigate this issue. For example, as shown in the additional experiments in our response to ***Reviewer qwJC Comment #1***, attribution methods that only focus on increasing the probability of the correct answer token (e.g. QRNCA) are more likely to be misled by correlated but non-causal neurons due to the collateral effect problem, while our cross-entropy-based scoring can better exclude those "fake" good/bad neurons by considering the contrastive information from both correct and wrong answers. We hope this analysis can provide useful insights for future works and look forward to a systematic investigation of this question.
> >
> > >**Comment #8:** Sensitivity to Distractor Quality.
> >
> > We appreciate this insightful question. Intuitively, the quality of distractor choices can affect the identification. To investigate this, we conduct additional experiments comparing two distractor generation methods for NeuronLLM on NER task with LlaMA 2-7B: random sampling from the possible answer space and LLM-generated distractors. The prompt for LLM generation is: "Here is a question: [question]. The correct answer is [correct answer]. Please generate 3 wrong options that are plausible but incorrect for it". The model we utilize to generate distractors is Gemini 2.5 Pro.
> >
> > **Table 6:** Compare the control performance using Random Sampling vs. LLM-generated distractors. Results show RAC/RCC metrics for degrade and enhance operations on NER task. The model here is LLaMA 2-7B.
> >
> > | Operation | Random Sampling | LLM-generated |
> > | :--- | :--- | :--- |
> > | degrade | 53.3/64.0 | 53.3/62.0 |
> > | enhance | 25.6/46.0 | 24.4/46.0 |
> >
> > As the results in Table 6 show, the control performance between using random distractors and LLM-generated distractors is very similar. This suggests that as long as the distractors are reasonably plausible, the specific generation method may not significantly impact the neuron identification and control effectiveness of our method. Note that this might be due to Gemini 2.5 Pro is already very strong in understanding the task and our intention so the generated distractors are of high-quality. Therefore we always recommend using strong LLMs for distractor generation when we don't know the possible answer space. Once we know the possible answer space, as the experiments in the main paper show, random sampling is often a reliable and cost-effective way to generate distractors. We will include these new results in the revised paper.

---

### Official Review · Reviewer_qwJC · 2025-11-03

**Soundness:** 1
**Presentation:** 3
**Contribution:** 1
**Rating:** 2
**Confidence:** 5

**Summary:**

This work proposes NeuronLLM, a novel framework that identifies both good and bad neurons in LLMs. NeuronLLM consists of two key modules: Answering-based Task Transformation (QATT) module and a Contrastive Neuron Identification (CNI) module.
(1) QATT. This module offers an effective way to transform diverse tasks to a universal multi-choice QA form
(2) CNI. This module adopts a cross-entropy-based contrastive neuron scoring method that is naturally suited for the QA format
Extensive results on LLaMA 2-7B, Baichuan 2-7B, and LLaMA 2-13B show that NeuronLLM substantially outperforms state-of-the-art methods over multiple NLP tasks

**Strengths:**

* this work is easy to follow and the motivation is clearly stated
* identifying both good and bad neurons is an interesting strategy

**Weaknesses:**

* this work is highly similar to an existing work, QR-Neuron [1], which severely undermines the novelty and contribution of this manuscript. (1) QR-Neuron is the first work that introduced multi-choice QA for neuron analyses, and this work claims that they propose this strategy and didn't give proper credit to prior work; (2) the proposed QATT also follows the core idea of the QR-Neuron work. I would suggest clarifying the distinction and novelty of QATT
* the authors claim that *"Cross-Entropy-based Contrastive Neuron Scoring can capture both the confidence of the LLM in the
correct choice and its uncertainty about incorrect ones."*  *"the probability of generating the correct choice over the whole vocabulary,
leading to wrongly identified neurons that actually increase/decrease both probabilities of correct and incorrect answers"*. I find this argument misleading. The probability is obtained by using softmax over the vocabulary, which also accounts for both correct and wrong answers.

[1] Identifying Query-Relevant Neurons in Large Language Models for Long-Form Texts. AAAI 2025

**Questions:**

* I find that *NeuronLLM* is not an appropriate name since this work aims to identify neurons in LLMs rather than developing a new LLM
* in line 261, does it mean that a proxy question set always consists of three questions? Why is the number fixed to three?
* it is useful to give a definition about *good* and *bad* neurons

---

> ### Author Response · Authors · 2025-11-22
> **response for comment #1**
>
> We sincerely appreciate your constructive comments. Please see our response to your comments one by one below.
>
> >**Comment #1:**  Clarifying the distinction and novelty.
>
> We thank the reviewer for this important feedback regarding QR-Neuron and offering us the clarification chance to avoid further misunderstanding. QR-Neuron is actually the "QRNCA" baseline in this paper which we compared against in our experiments, cited at multiple points (e.g., Section 4.1, Section 4.3) and mentioned throughout the paper (e.g. Table 1, Table 2, Table 6, Table 7). We're sorry that our current presentation may not have given enough credit to it, especially in the QATT section and we apologize for this oversight. However, the motivation of our QATT is to accommodate our Contrastive Neuron Identification (CNI) module: only when we have correct and wrong options, we can "contrastingly" identify neurons that can positively or negatively separate correct and wrong answers. Hence we designed QATT to provide such context. Besides this, there are also fundamental conceptual differences between our approach and theirs, although the multiple-choice format is indeed a common point. Due to the limited space in the main paper, we didn't elaborate on this baseline as much as we should have, and now we'll provide a more detailed clarification in this response.
>
> To better clarify the distinctions, we will first answer your second concern ("... The probability is obtained by using softmax over the vocabulary, which also accounts for both correct and wrong answers ...")
> , which is about the difference between our Cross-Entropy-based Contrastive Neuron Scoring and the probability of generating the correct choice over the whole vocabulary.
>
> ###  Conceptual Distinction: Attribution Objective Redefinition
>
> The core difference lies in **how we define the attribution objective**, i.e., the utility of neurons and the mechanism to achieve the objective:
>
> **QRNCA's approach:**
> - **Attribution Objective:** Increase $P(\text{correct answer})$, the answer is one token in $(A, B, C, D)$.
> - **Mechanism:** A neuron is deemed "good" if it increases the probability of the correct answer token (*e.g.*, $P(A)$). No matter whether it increases or decreases the probability of wrong options (*e.g.*, $P(B)$, $P(C)$ and $P(D)$). Note that since the probability is obtained by softmax over the whole vocabulary, $P(A)$, $P(B)$, $P(C)$ and $P(D)$ can be simultaneously increased by decreasing the probabilities of the other massive irrelevant tokens in the vocabulary. In the worst case, enhancement of the correct answer can lead to the probability of some wrong answers increase and surpass it; also, degradation of the correct answer can lead to the probabilities of all wrong answers decrease together where the model's ranking for the four options stays unchanged.
>
> **Our approach:**
> - **Attribution Objective:** The alignment between the model's prediction distribution over the correct and wrong options $(A,B,C,D)$ and the true label distribution.
> - **Mechanism:** A neuron is deemed "good" if it decreases cross-entropy between model prediction and true label over the four options.
>
> ### Why This Distinction Matters: The "Collateral Effect"
>
> This is not a trivial difference. In QRNCA, a vocabulary of 32,000 tokens (LlaMA 2-7B) offers a vast "buffer" space for the probability assignment of correct and wrong answers, which means when increasing or decreasing the probability of the correct answer, the probability of wrong answers can move in the same direction. Our additional experiments in Table 1 provide empirical evidence of such "collateral effect".
>
> **Table 1:** Comparison of QRNCA and our method, revealing the *"collateral effect"* of QRNCA. **Enhance(C/W)** and **Suppress(C/W)** show the average relative probability change for correct/wrong options after enhancement and suppression. For wrong options, we first average across the three incorrect options, then average across all test questions. Here we use 300 NER questions as the example. "+" indicates probability increase and "-" indicates decrease. **Collateral: x/y/z** reports collateral effect counts, where $z$ = #questions with correct option probability changed in the desired direction, $y$ = #questions where $\ge$1 wrong option also changed in the same direction as the correct option (mild collateral), and $x$ = #questions where all three wrong options changed in the same direction (severe collateral).
>
> | Method | Enhance (C/W) | Suppress (C/W) |
> | :--- | :--- | :--- |
> | **QRNCA** (whole vocabulary) | +138.9% / +106.2% (Collateral: 19/151/245) | -98.7% / -81.9% (Collateral: **272/300/300**) |
> | **Ours** (4 options; good neurons) | +155.1% / -22.9% (Collateral: 3/62/251) | -54.2% / +339.5% (Collateral: 81/197/284 ) |
> | **Ours** (4 options; good & bad) | +184.2% / -20.5% (Collateral: 2/40/244) | -53.0% / +636.6% (Collateral: 47/183/281) |

---

> > ### Author Response · Authors · 2025-11-22
> > **response for comment #1 (continued)**
> >
> > As we can see, the average statistical results tell us QRNCA is indeed more likely to exhibit the same-direction changes (both "+" for enhancement, both "-" for suppression). In contrast, our method effectively increases the correct option probability while decreasing the wrong ones for enhancement and vice-versa for suppression. Notably, the collateral effect for suppressing the QRNCA detected neurons is extremely severe: 272/300/300 means for all the 300 questions, at least one wrong option's probability drops simutaneously with the correct option, and 272 having all 4 options drop together (about 91% test questions), while our method substantially mitigates this issue by redefining the attribution target using crossen-tropy. Suprisingly, jointly editing both good and bad neurons further reduces collateral effect, highlighting the effectiveness of our approach compared to QRNCA.
> >
> > The above discussions provide conceptual explanation and empirical evidence regarding the collateral effect in QRNCA which is one key reason that makes QRNCA less effective than ours. Note that in some cases, the collateral effect itself might not be a problem. For example, when the probability of the correct and wrong options are simultaneously increased, if a neuron can increase the correct option probability more than the wrong ones, it can still be deemed as a good neuron. Because although all four options have increased probabilities, the gap between the correct and wrong options is enlarged, which is beneficial for model discrimination. The root problem is the gap between correct and wrong answers can either be enlarged or shrunk due to the collateral effect, making the identification of QRNCA less reliable. Therefore, we propose that a model's task ability is not solely related to the correct answer probability, but more importantly, how the model can differentiate between, which is exactly what cross-entropy measures. The superiority of this design is supported by our extensive experiments in the main paper.
> >
> > ### Innovation Beyond QRNCA
> >
> > In summary, although both our method and QRNCA use a multiple-choice QA format, the motivations are fundamentally different, and we believe our contributions go beyond this shared aspect:
> >
> > 1. **Conceptual:** Defining the attribution objective at a more precise option-level, utilizing the contrastive information from correct and wrong answers, mitigating the problems brought by the "collateral effect".
> > 2. **Methodological:** Identifying and simultaneously controlling *both* good and bad neurons yield better control, as supported by all experiments in our main paper.
> > 3. **Empirical:** Demonstrating the effectiveness and superiority of this approach with extensive experiments.

---

> > > ### Author Response · Authors · 2025-11-22
> > > **response for comments #2~4**
> > >
> > > >**Comment #2:** I find that NeuronLLM is not an appropriate name since this work aims to identify neurons in LLMs rather than developing a new LLM.
> > >
> > > We will rename our method to better reflect its purpose.
> > >
> > > >**Comment #3:** In line 261, does it mean that a proxy question set always consists of three questions? Why is the number fixed to three?
> > >
> > > The reason to have multiple proxy questions is to avoid LLMs sometimes answering questions correctly by chance which can mislead our neuron attribution. Having more proxy questions can reduce this risk but also increase computational cost. We empirically found three questions can already result in a good performance while keeping the computational cost manageable. We will clarify this in the revised paper.
> > >
> > > >**Comment #4:** It is useful to give a definition about good and bad neurons.
> > >
> > > The definitions of good and bad neurons are provided in Section 3.4. Essentially, based on the ACE scores shown in formula (5), good neurons are the top-K neurons with the highest ACE scores, while bad neurons are the top-K neurons with the lowest ACE scores. Good neurons mean that boosting/suppressing them can positively/negatively contribute to the attribution target (decrease/increase cross-entropy), while bad neurons mean that suppressing/boosting them can positively/negatively contribute to it (decrease/increase cross-entropy). We will make this clearer in the revised paper.
> > >
> > > We thank the reviewer for these helpful comments. We will revise the paper to clearly acknowledge QRNCA's pioneering contribution while articulating our specific innovations and address all the questions raised in our revised paper. We hope this clarification addresses your concerns.

---

### Note · Authors · 2026-01-05

I have read and agree with the venue's withdrawal policy on behalf of myself and my co-authors.